# Transcriptomic Profile Analysis of *Populus talassica* × *Populus euphratica* Response and Tolerance under Salt Stress Conditions

**DOI:** 10.3390/genes13061032

**Published:** 2022-06-08

**Authors:** Ying Liu, Zhan Jiang Han, Meng Xu Su, Min Zhang

**Affiliations:** 1College of Life Science and Technology, Tarim University, Alar 843300, China; liuying836354822@126.com (Y.L.); sumengxu403@163.com (M.X.S.); zm1261262022@126.com (M.Z.); 2Xinjiang Production and Construction Corps Key Laboratory of Protection and Utilization of Biological Resources in Tarim Basin, Alar 843300, China

**Keywords:** *Populus talassica* *× Populus euphratica*, salt stress, transcriptome sequencing, differentially expressed genes

## Abstract

A new *Populus* variety with a strong salt tolerance was obtained from cross breeding *P. talassica* as the female parent and *P. euphratica* as the male parent. In order to elucidate the molecular mechanism and find out the major differentially expressed genes of salt tolerance of *P. talassica × P. euphratica*, after being subjected to salt stress, at 0, 200, and 400 mmol/L NaCl, the root, stem, and leaf transcriptomes (denoted as R0, S0, and L0; R200, S200, and L200; and R400, S400, and L400, respectively) of *P. talassica* × *P. euphratica* were sequenced. In total, 41,617 differentially expressed genes (DEGs) were identified in all the comparison groups with 21,603 differentially upregulated genes and 20,014 differentially downregulated genes. Gene Ontology analysis showed that DEGs were significantly enriched in biological processes that may be involved in salt stress, such as ‘cell communication’, ‘ion transport’, ‘signaling’, and signal ‘transmission’. Kyoto Encyclopedia of Genes and Genomes analysis showed that DEGs were mainly enriched in pathways of ‘plant–pathogen interaction’, ‘carbon metabolism’, and ‘plant hormone signal transmission’. The pathways and related gene information formed a basis for future research on the mechanisms of salt stress, the development of molecular markers, and the cloning of key genes in *P. talassica* × *P. euphratica*.

## 1. Introduction

Soil salinization is a main challenge hindering global food security and environmental sustainability. The harmful effects of climate change have accelerated the development of soil salinity [1]. Low precipitation, high surface evaporation, weathering of primary rocks, and irrigation with brine are the main reasons for the increase in salinity [2]. Soil salinization poses multiple hazards to crops. Salt stress causes physiological and metabolic crop disorders, such as decreased photosynthetic efficiency, increased respiration, blocked protein synthesis, accumulation of toxic substances, and accelerated aging and death [3]. Due to frequent drought, irregular precipitation, increasing salinization, and high temperature, climate change has altered many ecosystems. These environmental changes have also led to a decline in crop yields worldwide. Therefore, there is an urgent need to fully understand the response mechanisms of plants to abiotic stress and use the acquired knowledge to improve their stress resistance [4].

Saline soil mainly contains Na^+^ and Cl^−^ [5]. Plants grown in saline soil can regulate the absorption of Na^+^ and Cl^−^ to avoid ionic toxicity and ensure that there is enough solute to regulate permeability [6]. To survive in these areas, plants need to produce complex responses to these abiotic stressors, including signal transduction, gene expression regulation, ion homeostasis, reactive oxygen species clearance, compatible solute accumulation, and growth regulation [7,8,9,10,11]. Plants produce large numbers of transduction signals under salt stress conditions, including ion, osmotic, and detoxification signals. In addition, the sucrose and starch catabolic process in plants is strengthened. The decomposed sugars provide energy for plant growth and development, and the resulting small molecular monosaccharides increase the soluble sugar contents of cells and the osmotic potential of cells, which aid in resisting the osmotic stress caused by high salt ion concentrations [12]. In crops, salt not only affects osmotic stress, it also affects ion stress, and secondary stresses, such as oxidative stress, also occur [13].

Abiotic stresses trigger a series of plant responses, starting from stress perception, leading to the activation of signal pathways and changes in gene expression levels, thereby altering plant physiology, growth, and development [14]. Salt stress affects all the stages of plant growth and development (such as seed germination, growth, flowering, and fruiting) and all the processes of physiological metabolism (such as water metabolism, photosynthesis, and enzyme system metabolism), which leads to declines in crop yield and quality, and even crop failure in serious cases. The responses of plants to salt stress include a series of changes at the molecular, biochemical, and physiological levels. Salt stress leads to the destruction of intracellular homeostasis and ion distribution, as well as the denaturation of structural and functional proteins [15]. To improve the salt tolerance of plants, it is necessary to understand the salt tolerance mechanisms of plant growth and physiology, as well as the salt tolerance mechanisms at the whole-plant, organelle, and molecular levels [16]. Under salt stress conditions, the gene expression pattern changes, and special genes related to salt stress are activated [15]. Identifying genes that enable plants to adapt to, or tolerate, salt stress is key for breeding programs. It is particularly urgent and important to improve the utilization rates of salinized land resources, ensure ecological security, and achieve high quality crops with high yields. This may be achieved by studying the mechanisms of plant salt tolerance and cultivating salt-tolerant plant varieties.

To develop more salt-tolerant crops effectively, the genetic mechanisms of salt tolerance must be determined. Salt stress changes gene expression in tissue-specific and time-dependent manners. Identifying transcripts associated with salt stress is the first step in detecting genes that contribute to plant salt tolerance. The structural analysis of isolated genes will provide clues to the elements controlling their expression [5]. Identifying the key genes involved in the reproductive physiology of halophytes and using them to transform crops is a promising method for the development of saline agriculture [17]. Previous studies have shown that plant salt tolerance involves multiple genes and is coordinated by multiple mechanisms [18]. The genes related to salt stress response involve ion transport, cell defense, physiological metabolism, cell growth, and many other pathways. These genes work together to resist salt stress in different ways, such as encoding genes related to photosynthesis, osmotic regulation, free radical scavenging enzymes, and vacuole regionalizing enzymes [19].

The use of salinized land for crop cultivation and production has become an important factor for agricultural development [3,19]. Improving the utilization of saline alkali land requires the application of biological improvement methods to increase land-use efficiency [20]. To date, numerous genetic-improvement-related investigations of crop plants have been conducted to elucidate changes in species at the transcriptome level in response to salt stress. The studied plants include *Arabidopsis* [21], rice [22], peanut [23], sugar beet [24], cotton [25], citrus [26], *Clerodendrum inerme* (L.) [27], Microalgae *Dunaliella* [28], kenaf [29], and wild barley [30].

*P. talassica* × *P. euphratica* is a new variety obtained by cross breeding *P. talassica* Kom. as the female parent and *P. euphratica* Oliv. as the male parent. In addition to the excellent characteristics of cold resistance, drought resistance, saline alkali resistance, barren resistance, and wind and sand resistance for *P. talassica* × *P. euphratica*, it also has the characteristics of strong asexual reproductive ability, fast growth speed, strong growth potential, good material quality, high cutting survival rate, and strong adaptability. It is an excellent tree species for wind prevention and sand control afforestation in arid, saline alkali, and desert areas. At present, although some research progress has been made in salt-tolerance-related gene mining and in understanding the molecular mechanisms of salt tolerance in other crops, research on the salt tolerance of *P. talassica* × *P. euphratica* remains at the morphological and physiological levels. The related molecular mechanisms are still unclear, and there is limited research on transcriptome analysis of different tissues in *P. talassica* × *P. euphratica* under salt stress conditions. In recent years, high-throughput sequencing technologies have been used to study gene function and gene structure at the overall level, and it has revealed the molecular mechanisms of specific biological and physiological processes. Among these technologies, a transcriptome sequencing (RNA-seq) is a collection of all the transcripts produced by a species or specific cell type, and it has been widely used in many fields, such as basic research, physiological mechanisms, and clinical diagnoses [31]. Therefore, RNA-seq is a cost-effective platform for analyzing gene expression [32,33,34]. However, the differences in the salt tolerance mechanisms of *P. talassica* × *P. euphratica* and the elucidation of genes that play major roles in these mechanisms are limited, due to the lack of transcriptome information.

To explore the unique salt tolerance mechanisms of *P. talassica* × *P. euphratica* and explore its key genes in response to salt stress, this study applied RNA-seq technology for root, stem, and leaf tissues of *P. talassica* × *P. euphratica* under salt stress conditions. After transcriptome analysis of *P. talassica* × *P. euphratica* under salt stress conditions were performed, the differentially expressed genes under these conditions were excavated, and pathways related to the salt stress responses of *P. talassica* × *P. euphratica* were identified. Understanding the salt tolerance mechanisms of *P. talassica* × *P. euphratica* at the molecular level will not only provide a valuable resource for future research and breeding [35] but will also help to promote the planting of *P. talassica* × *P. euphratica* in saline alkali land and provide a reference for the improvement and sustainable development and utilization of soil saline alkali land in Xinjiang in the future.

## 2. Materials and Methods

### 2.1. Plants and Stress Treatment

One-year-old potted *P. talassica* × *P. euphratica* seedlings were used as experimental materials to study the salt tolerance of *P. talassica* × *P. euphratica* at different NaCl concentrations by open-air potted soil culture. Three NaCl concentration levels were used—0, 200, and 400 mmol/L. *P. talassica* × *P. euphratica* with the same growth trends and good growth were selected and they were divided into three groups of three plants each. The salt treatment group was treated with NaCl solution once every three days, and the control group was irrigated with 1000 mL deionized water once every three days. Both the control and the salt treatment groups were treated for a total of five times.

In order to avoid the salt shock effect, the potted *P. talassica* × *P. euphratica* seedlings were treated with salt stress in the way of increasing salt concentration until the predetermined concentration. In the 200 mmol/L NaCl treatment group, NaCl concentration increased by 50 mmol/L, 100 mmol/L, 150 mmol/L, and 200 mmol/L until the predetermined salt concentration of 200 mmol/L, and then were treated with 200 mmol/L NaCl solution once. We kept the final salt concentration stable at 200 mmol/L. In the 400 mmol/L NaCl treatment group, NaCl concentration increased by 100 mmol/L, 200 mmol/L, 300 mmol/L, and 400 mmol/L, until the predetermined concentration of 400 mmol/L, and then was treated with 400 mmol/L NaCl solution once. We kept the final salt concentration stable at 400 mmol/L. After the salt concentration was stabilized, the roots, stems, and leaves of seedlings in each treatment group were selected, with a total of 27 samples. After sampling, samples were quickly frozen in liquid nitrogen and brought back to the laboratory for storage in a −80 °C ultra-low-temperature refrigerator.

### 2.2. RNA Extraction, Library Construction, and Sequencing

The root, stem, and leaf samples treated with 0, 200, and 400 mmol/L NaCl solution were set as R0, S0, and L0, respectively; R200, S200, and L200, respectively; and R400, S400, and L400, respectively. RNA was extracted from tissues or cells using the TRIzol extraction method, followed by the rigorous quality control of the RNA samples using an Agilent 2100 BioAnalyzer and the accurate detection of RNA integrity.

The first-strand cDNA was synthesized using an M-MuLV reverse transcriptase system with an mRNA fragment as template and a random oligonucleotide as primer. Subsequently, the RNA strand was degraded by RNaseH, and the second-strand cDNA was synthesized with dNTPs in a DNA Polymerase I system. The purified double-stranded cDNA was repaired at the end, A tails were added, and sequencing joints were connected. Approximately 250–300-bp cDNA was screened using AMPure XP Beads for PCR amplification, and PCR products were purified again with AMPure XP beads to finally obtain libraries [36].

After the library was constructed, a Qubit2.0 Fluorometer was used for initial quantification, and the library was diluted to 1.5 ng/µL. An Agilent 2100 BioAnalyzer was then used to determine the insert size of the library, and qRT-PCR was used to accurately quantify the effective library concentration (Effective library concentration is higher than 2 nM) to ensure the library quality. After qualified library inspection, Illumina sequencing was performed using Illumina HiSeq 2500 after pooling different libraries in accordance with the effective concentration and the targeted offline data requirements. The basic principle of this sequencing technology was sequencing by synthesis. In the sequencing flow cell, there were four kinds of fluorescence-labeled dNTPs, amplification primers, and DNA polymerase. In each sequence complementary chain cluster, when a fluorescently labeled dNTP was added to the chain, it released the corresponding fluorescence, which was captured by Illumina HiSeq 2500 and converted from optical signals into sequencing peaks by computer software. In this manner, the sequence information of the fragment to be tested was obtained [37,38,39].

### 2.3. Data Quality Control and Reference Genome Alignment

The original data obtained by sequencing contained a small number of reads contaminated with sequencing connectors or of low sequencing quality. To ensure the quality and reliability of the data analysis, it was necessary to filter the original data, remove the reads with adapters, and remove the reads containing large numbers of Ns (where N indicates that the base information cannot be determined). Low-quality reads (Qphred ≤ 20 base numbers represent more than 50% of the total read length) were removed. Clean reads for the subsequent analysis were obtained by filtering the original data filtering and confirming the sequencing error rate and GC content distribution [40]. Clean reads after quality control were compared with the reference genome. HISAT2 (Version 2.0.4) software (http://daehwankimlab.github.io/hisat2/ (accessed on 3 October 2021)) was used to compare clean reads quickly and accurately with the reference genome of *P. talassica × P. euphratica* to obtain the locational information of reads on the reference genome [41].

### 2.4. Quantification of Gene Expression and Differential Expression Analysis

Using the locational information of the gene alignment on the reference genome of *P. talassica × P. euphratica*, the number of reads covering each gene (including the new prediction gene) from start to end was calculated. Reads with a comparison quality value of less than 10, unpaired reads, and reads to multiple regions of the genome were filtered. This part of the analysis was performed using the featureCounts tool in subread software [42]. A quantitative analysis of the gene expression level was carried out for each sample, and then the expression matrix of all the samples was obtained [43].

Due to the influence of sequencing depth and gene length, RNA-seq expression values are generally not expressed by read count, but by FPKM, which is used to correct sequencing depth and gene length successively [44]. After the expression values of all the genes in each sample were calculated, the distributions of gene expression levels in different samples were displayed using box graphs. After gene expression quantification, the expression data were statistically analyzed to screen for genes with significant differences in expression level under different conditions. The difference analysis was divided into three steps. The original read count was first standardized, mainly to correct for the sequencing depth. Then, the statistical model calculated the probability of hypothesis testing (*p*-value), and finally, multiple hypothesis testing correction was conducted to obtain false discovery rate values (error detection rate, padj) [45,46,47].

Transcriptome analyses are conducted on thousands of genes, which leads to the accumulation of false positives. The greater the number of genes, the greater the accumulation of false positives in the hypothesis testing. Therefore, padj was introduced to correct the *p*-value of the hypothesis testing to control the proportion of false positives [48]. Differences in gene screening standards are very important, we used the standard |log_2_ (fold change)| ≥ 1 and padj ≤ 0.05 as common values.

### 2.5. Novel Gene Prediction

In order to identify possible key genes related to salt resistance of *P. talassica × P. euphratica*, we used |log_2_ (fold change)| ≥ 5 as a selection criterion. Some genes with high expression levels were screened out from the list of differentially expressed genes (DEGs) in all comparison groups of root, stem, and leaf.

### 2.6. Gene Ontology (GO) Enrichment Analysis and Kyoto Encyclopedia of Genes and Genomes (KEGG) Pathway Enrichment Analysis

Gene ontology uses a comprehensive database to describe the functions of genes. It mainly classifies genes on the basis of their essential functions to define and describe the functions of genes and proteins. The GO database divides the functions into three types: Biological Process (BP), Cellular Component (CC), and Molecular Function (MF) [49,50,51]. The KEGG database integrates genomic, chemical, and system functional information [52,53].

Enrichment analysis was based on the principle of hypergeometric distribution, in which the differential gene set was the gene set obtained from the differential significant analysis and annotated in the GO or KEGG database. ClusterProfile software was used for the GO functional enrichment analysis of differential gene sets. In addition, ClusterProfile software was used to analyze the KEGG pathway enrichment of differential gene sets. The KEGG database is comprehensive, integrating genomic, chemical, and system functional information. Both GO and KEGG enrichment analyses used padj <0.05 as the threshold for significant enrichment [54]. In BP, enrichment was most concentrated in cellular processes and metabolic processes. Cells and cell components were the most enriched in CC. Binding and catalytic activities were significantly enriched in MF pathways. KEGG provides a way to systematically analyze gene function according to the metabolic network of gene products [55].

### 2.7. Mutation Locus Analysis

A mutation site analysis is an important RNA-seq structural analysis, mainly including the detection of congenital and acquired somatic mutation sites. GATK software (https://gatk.broadinstitute.org (accessed on 3 October 2021)) was used to analyze the variation sites in the sample data, and SnpEff software was used to annotate the variation sites [56].

## 3. Results

### 3.1. Sequencing Quality Analysis

To ensure the quality and reliability of the data analysis, first, the original data were filtered, and then the reads with joints, reads containing N, and low-quality reads were removed. After original data filtering, the sequencing error rate and GC content distribution were determined. In total, 1,109,438,718 clean reads were obtained, accounting for 97.08% of the raw reads. The data summary is shown in Table 1. In 27 samples, the proportion of the high-quality read content to the original read content was greater than 94%. There was 166.41 G of high-quality reads, and the content of GC per sample was greater than 44%. Thus, the reads were of high quality. High quality reads with Q20 exceeded 97.31%, and those with Q30 exceeded 92.5%. The GC contents in both groups exceeded 42.81%. These data results revealed the sequencing quality of the transcriptome. Among the 27 samples, high-quality reads (high Q20 percentage) were selected. 

### 3.2. Reference Genome Alignment Results Were Analyzed

Clean reads after quality control were compared with the reference genome of *P. euphratica*. HISAT2 (Version 2.0.4) software (http://daehwankimlab.github.io/hisat2/ (accessed on 3 October 2021)) was used to compare clean reads quickly and accurately with the reference genome to obtain the locational information of the reads on the reference genome. To calculate the respective mapping rates of read1 and read2, the Total_reads number was the sum of read1 and read2, which was Clean_reads in Table 1, and the comparison between the samples and reference genomes is shown in Table 2.

### 3.3. Quantitative Analysis of Gene Expression

#### 3.3.1. Gene Expression Distribution

Using the locational information from the gene alignment on the reference genome of *P. talassica* × *P. euphratica*, the number of reads covering each gene (including the newly predicted genes) from start to end was calculated. Reads with a comparison quality value of less than 10, unpaired reads, and reads to multiple regions of the genome were filtered. This part of the analysis used the featureCounts tool in the subread software [42]. 

Then, HISAT2 (Version 2.0.4) software (http://daehwankimlab.github.io/hisat2/ (accessed on 3 October 2021)) was used to compare the obtained high-quality clean reads to the reference genome of *P. talassica* × *P. euphratica*, and StringTie software (https://ccb.jhu.edu/software/stringtie/ (accessed on 3 October 2021)) was used to perform a quantitative expression analysis of each sample gene [57]. After all the gene expression values (FPKM) of the 27 samples were calculated, the distribution of gene expression levels among samples was determined, as shown in Figure 1. The gene expression distribution of all samples in each treatment group of root, stem, and leaf can be clearly seen from Figure 1A.

#### 3.3.2. FPKM Density Distribution

Generally speaking, the number of DEGs only accounts for a small part of the whole gene, and a few differentially expressed genes have little influence on the distribution of the expression level of a sample. In most cases, samples have similar distributions of expression levels. For the gene expression level of each sample, the logarithm base 2 was used to construct the density distribution map, as shown in Figure 1B. The abscissa represents log_2_ (FPKM + 1) values, and the ordinate represents the gene distribution density of the corresponding expression quantity. Thus, they reflect the distribution of gene FPKM and the proportions of genes with different expression levels in all the samples. Each color in the figure represents a sample, the sum of all probabilities is 1, and the area of each region is 1. The peak of the density curve represents the region with the highest concentration of gene expression in the whole sample. In the data, there were different peaks, but the proportions of gene expression at different peaks and the trend of density distribution curves of different samples were relatively consistent because the distribution of the gene expression density in the same tissue of the same species was similar [58].

#### 3.3.3. Correlation Analysis between Samples

Inter-sample correlation, intragroup biological repeatability, and intergroup sample differences were evaluated. The correlation of gene expression levels among samples represents an important index to determine the reliability of experiments and the rationality of sample selection. The closer the correlation coefficient is to 1, the greater the similarity in the expression patterns between samples. In accordance with the FPKM values of all the genes in each sample, the correlation coefficients of samples within and between groups were calculated, and a heat map was constructed that can visually display the sample differences between groups and the sample repetition within groups. The higher the correlation coefficient between samples, the closer the expression pattern. The sample correlation heat map is shown in Figure 1C. The left and upper sides are sample clustering, the right and lower sides are sample names, and the squares of different colors represent the correlations of the two samples. In Figure 1C, the correlation coefficient of the three biological replicates for each salt concentration treatment was close to 1, which showed that the data of the three biological replicates at each salt treatment were very good, and the correlations within and between groups were good. Thus, they can be used for subsequent experiments.

A principal component analysis (PCA) is an algorithm that reduces data dimensions. It is a way to evaluate the quality of biological duplications. It is used to evaluate the biological repeatability of samples within a group and the differences between groups. On the basis of a data matrix of gene expression in the sample, it is mapped to the two-dimensional plane to obtain the principal component (PC) with the greatest contribution to the different genes. In the transcriptome, each sample contains thousands of genes. The FPKM values of these genes were calculated, and a PCA was used to reduce the dimension of the overall data. The distance between the midpoint in Figure 1D represents the degree of similarity between samples. PC1 refers to the contribution rate ranking first, which is the factor that has the greatest impact on the amount of variation. PC2 is the factor ranking second. As shown in Figure 1D, the abscissa was PC1, and the ordinate was PC2. As shown in the PCA cluster diagram of Figure 1D, the correlations of the three biological repeats in the R0 and L0 treatment groups were low, and the difference between groups was obvious. The three biological repeats in other groups, such as L200, L400, S0, S200, S400, R200, and R400, gathered together. The PC1 and PC2 of the three biological repeats were very close, the correlations were high, and the difference between groups was not obvious, indicating that the samples were very similar.

### 3.4. Analysis of Differential Gene Expression Levels

#### 3.4.1. DEGs in Different Comparison Groups

On the basis of the expression of quantitative data, using the DESeq2 identification of differentially expressed genes, poplar seedlings in different experiments were compared. The screening criteria for DEGs were |log_2_ (fold change)| ≥ 1 and padj ≤ 0.05. When *p* ≤ 0.05, the gene was regarded as differentially expressed. In Figure 2, the abscissa represents the comparison groups in the differential expression analysis, the ordinate represents the number of DEGs, and the color represents upregulated and downregulated genes. As shown in Figure 2, the numbers of upregulated and downregulated genes in each comparison group of *P. talassica* × *P. euphratica* roots, stems, and leaves were consistent with volcano diagram results. As shown in Figure 2, under salt stress conditions, there were more upregulated genes than downregulated genes, indicating that the effect of salt stress on the gene expression trend was mainly upregulated.

#### 3.4.2. Differential Gene Volcano Map

In Figure 3A–I, the abscissa is log_2_(fold change), the ordinate is the significance level, and the value of negative logarithm was taken for 10. The two vertical dashed lines in the figure represent the threshold of the difference multiple, and the horizontal dotted line is the significance level threshold. Color indicates that the gene is upregulated, downregulated, or non-significantly differentially expressed. The downregulated genes are represented in green, the upregulated genes are represented in red, and the non-differentially expressed genes are represented in blue.

##### Volcano Map of Differential Gene Expression Levels in Leaf Comparison Groups

The levels of differential gene expression are shown using a volcano map. As shown in Figure 3A–C, the total number of DEGs in the L200 vs. L0 comparison group was 3042, with 1842 being upregulated and 1200 being downregulated. The total number of DEGs in the L400 vs. L0 comparison group was 4534, with 2472 being upregulated and 2062 being downregulated. The total number of DEGs in the L400 vs. L200 comparison group was 4250, with 2197 being upregulated and 2053 being downregulated.

##### Volcano Map of Differential Gene Expression Levels in Stem Comparison Groups

The levels of differential gene expression were shown using a volcano map. As shown in Figure 3D–F, the total number of DEGs in the S200 vs. S0 comparison group was 8257, with 4564 being upregulated and 3693 being downregulated. The total number of DEGs in the S400 vs. S0 comparison group was 8478, with 4562 being upregulated and 3916 being downregulated. The total number of DEGs in the S400 vs. S200 comparison group was 1548, with 626 being upregulated and 922 being downregulated.

##### Volcanic Map of Differential Gene Expression Levels in Root Comparison Groups

The levels of differential gene expression were shown using a volcano map. As shown in Figure 3G–I, the total number of DEGs in the R200 vs. R0 comparison group was 4076, including 1804 upregulated DEGs and 2272 downregulated DEGs. The total number of DEGs in the R400 vs. R0 comparison group was 5047, with 2291 being upregulated and 2756 being downregulated. The total number of DEGs in the R400 vs. R200 comparison group was 2385, with 1245 being upregulated and 1140 being downregulated.

#### 3.4.3. Differential Gene Venn Diagram

A Venn diagram of the co-expressed genes in each of the three comparison groups was constructed, and FPKM > 1 was used as the criterion to judge the gene expression. A Venn analysis was performed to detect the different expressions of rhizome and leaf tissues under salt stress conditions. The analysis results are shown in Figure 4A–F. In total, 271 genes were co-expressed in the R200 vs. R0, S200 vs. S0, and L200 vs. L0 comparison groups; 450 genes were co-expressed in the S400 vs. S0, L400 vs. L0, and R400 vs. R0 comparison groups; 38 genes were co-expressed in the R400 vs. R200, L400 vs. L200, and S400 vs. S200 comparison groups; 196 genes were co-expressed in the R200 vs. R0, R400 vs. R200, and R400 vs. R0 comparison groups; 381 genes were co-expressed in the S400 vs. S0, S200 vs. S0, and S400 vs. S200 comparison groups; and 150 genes were co-expressed in the L400 vs. L200, L200 vs. L0, and L400 vs. L0 comparison groups. In a comprehensive analysis, there were more DEGs in the Figure 4B,E comparison groups.

#### 3.4.4. Cluster Analysis of DEGs

The DEGs in all the comparison groups of roots, stems, and leaves were collected as the differential gene sets and clustered on the basis of their expression levels. In the cluster analysis of the differential gene sets, genes or samples with similar expression patterns in the heat map clustered together. Figure 5A shows the heat map of clustering among samples. The abscissa has the sample names, and the ordinate represents the normalized values of differential gene FPKM. Figure 5B shows an inter-group clustering heat map. Horizontal comparisons of colors in the heat map indicate the expression of the same gene in different samples. The higher the redness, the higher the expression level, and the higher the greenness, the lower the expression level. The Figure 5A,B shows the clustering of DEGs in each comparison group.

#### 3.4.5. Trend Analysis

To further provide a global expression profile of uniquely assembled transcripts under salt stress conditions, expression models of all the DEGs were created and divided into four clusters on the basis of a log_2_(fold change). Cluster is the cluster number after clustering according to the expression mode. As shown in Figure 6, the change trend of the expression of all the genes related to stress treatment was analyzed. The change characteristics of the same gene in a changing trend were determined to identify the most representative change process of a genetic group, as shown in Figure 6. Each square represents a kind of trend in the gene expression data. In Figure 6, the abscissa has the names of the different tissue samples, and the ordinate represents the value under the sample after Z-score homogenization of gene FPKM value in the sample. The gray line in each subplot represents the relative corrected gene expression levels of genes in a cluster under different experimental conditions, and and the blue line represents the average value of the relative corrected gene expression levels of all genes in this cluster under different experimental conditions. As shown in Figure 6, we can see the change trend of all gene expression levels after relative correction under different experimental conditions. Comprehensive analysis, it can be seen that the expression of most DEGs in sub_cluster (1–4) were up-regulated after salt stress exposure.

#### 3.4.6. Novel Gene Prediction Results

Using |log_2_ (fold change)| ≥ 5 as a selection criteria, by screening the DEGs in the three comparison groups of L200 vs. L0, L400 vs. L0, and L400 vs. L200, we found some genes that were highly expressed in leaves. In the L200 vs. L0 comparison group, the genes with higher expression levels were *LOC105129452*, *LOC105136804*, *LOC105127881*, *LOC105142149*, and *LOC105131213*. In the L400 vs. L0 comparison group, the genes with higher expression levels were *LOC105124563*, *LOC105129935*, *LOC105115176*, *LOC105141220*, and *LOC105132264*. In the L400 vs. L200 comparison group, the genes with higher expression levels were *LOC105128502*, *LOC105131234*, *LOC105124001*, *LOC105114274*, and *LOC105116811*.

Using |log_2_ (fold change)| ≥ 5 as a selection criteria, by screening the DEGs in the three comparison groups of S200 vs. S0, S400 vs. S0, and S400 vs. S200, we found some genes that were highly expressed in stem. In the S200 vs. S0 comparison group, the genes with higher expression levels were *LOC105141088*, *LOC105141220*, *LOC105119414*, *LOC105142033*, and *LOC105119522*. In the S400 vs. S0 comparison group, the genes with higher expression levels were *LOC105112164*, *LOC105129607*, *LOC105137881*, *LOC105126640*, and *LOC105131256*. In the S400 vs. S200 comparison group, the genes with higher expression levels were *LOC105116962*, *LOC105122741*, *LOC105115491*, *LOC105134677*, and *LOC105134493*. 

Using |log_2_ (fold change)| ≥ 5 as a selection criteria, by screening the DEGs in the three comparison groups of R200 vs. R0, R400 vs. R0, and R400 vs. R200, we found some genes that were highly expressed in roots. In the R200 vs. R0 comparison group, the genes with higher expression levels were *LOC105108844*, *LOC105135115*, *LOC105134637*, *LOC105115176*, and *LOC105114369*. In the R400 vs. R0 comparison group, the genes with higher expression levels were *LOC105127231*, *LOC105107217*, *LOC105124330*, *LOC105115119*, and *LOC105128502*. In the R400 vs. R200 comparison group, the genes with higher expression levels were *LOC105120059*, *LOC105116469*, *LOC105127348*, *LOC105112947*, and *LOC105124801*.

### 3.5. GO Enrichment Analysis of DEGs

To further analyze the expression functions of DEGs, a GO analysis was performed on all comparison groups of *P. talassica* × *P. euphratica* treated with different concentrations of NaCl, and the enriched GO items were screened on the basis of the significance of the enrichment (*p*-value ≤ 0.05) and reliability (Q-value ≤ 0.05). Each group of DEGs was significantly enriched at three levels: CC, BP, and MF. The GO functional enrichment used padj < 0.05 as the threshold of significant enrichment, and the GO enrichment results of all the comparison groups of *P. talassica* × *P. euphratica* roots, stems, and leaves are shown in Figure 7A–I. In the bar chart, the abscissa represents the GO terms, and the ordinate represents GO terms enriched −log_10_ (*p*-value).

#### 3.5.1. GO Enrichment Analysis Diagram of DEGs in Each Leaf Comparison Group

As shown in Figure 7A, the differentially expressed genes in the L200 vs. L0 comparison group were mainly enriched in cell communication, ion transport, cellular response to stimulus, signaling, and signal transmission in BP. Organelle part, intracellular organelle part, intracellular non-membrane-bound organelle, non-membrane bound organelle, and cell peripheries were mainly enriched in CC. Transferase activity, transferring glycosyl groups, transferring hexosyl groups, DNA-binding transcription factor activity, coenzyme binding, tetrapyrrole binding, and heme binding were mainly enriched in MF.

As shown in Figure 7B, the differentially expressed genes in the L400 vs. L0 comparison group were mainly enriched in ion transport, lipid metabolic process, cell communication, cellular response to stimulus, response to stress, signal transmission, signaling, oxidative metabolic process, and organic acid metabolic process in BP. Organelle part, intracellular organelle part, cell perimeter, membrane protein complex, non-membrane-bound organelle, and intracellular non-membrane-bound organelle were mainly enriched in CC. DNA-binding transcription factor activity, hydrolase activity, acting on glycosyl bonds, protein dimerization activity, transferase activity, transferring glycosyl groups, hydrolase activity, hydrolyzing O-glycosyl compounds, hydrolase activity, acting on acid anhydrides, in phosphorus containing anhydrides and hydrogen activity, and acting on acid anhydrides were mainly enriched in MF.

As shown in Figure 7C, the DEGs in the L400 vs. L200 comparison group were mainly enriched in ion transport, cellular response to stimulus, cell communication, lipid metabolic process, oxidative metabolic process, organic acid metabolic process, and carboxylic acid metabolic process in BP. Intracellular non-membrane-bound organelle, non-membrane-bounded organelle, intracellular organelle part, organelle part, membrane protein complex, catalytic complex, thylakoid, thylakoid part, and cell perimeter were mainly enriched in CC. DNA-binding, transcription factor activity, hydrolase activity, acting on glycosyl bonds, transferase activity, transferring glycosyl groups, hydrolase activity, hydrolyzing O-glycosyl compounds, protein dimerization activity, coenzyme binding, tetrapyrrole binding, and heme binding were mainly enriched in MF.

#### 3.5.2. GO Enrichment Analysis Diagram of DEGs in Each Stem Comparison Group

As shown in Figure 7D, the DEGs in the S200 vs. S0 comparison group were mainly enriched in cell communication, cellular response to stimulus, ion transport, lipid metabolic process, signal transduction, signaling, organic acid metabolic process, oxoacid metabolic process, carboxylic acid metabolic process, organophosphate metabolic process, and small molecule biosynthetic process in BP. Organelle part, intracellular organelle part, non-membrane-bound organelle, non-membrane-bound organelle, cell periphery, membrane protein complex, ribonucleoprotein complex, and catalytic complex were mainly enriched in CC. Transferase activity, transferring glycosyl groups, DNA binding transcription factor activity, transferring hexosyl groups, hydrolase activity, acting on acid anhydrides, hydrolase activity, acting on acid anhydrides, in phosphorus-containing anhydrides, pyrophosphatase activity, nucleoside-triphosphatase activity, hydrolase activity, acting on glycosyl bonds, and coenzyme binding were mainly enriched in MF.

As shown in Figure 7E, the DEGs in the S400 vs. S0 comparison group were mainly enriched in cell communication, lipid metabolic process, cellular response to stimulus, organic acid metabolic process, oxoacid metabolic process, carboxylic acid metabolic process, ion transport, signal transduction, signaling, and response to stress in BP. Organelle part, intracellular organelle part, non-membrane-bound organelle, intracellular non-membrane-bound organelle, cell periphery, and catalytic complex were mainly enriched in CC. Transferase activity, transferring glycosyl groups, transferase activity, transferring hexosyl groups, hydrolase activity, acting on glycosyl bonds, acting on acid anhydrides, in phosphorus-containing anhydrides, hydrolyzing O-glycosyl compounds, pyrophosphatase activity, and nucleoside-triphosphatase activity were mainly enriched in MF.

As shown in Figure 7F, the DEGs in the S400 vs. S200 comparison group were mainly enriched in ion transport, carboxylic acid metabolic process, oxoacid metabolic process, organic acid metabolic process, and catabolic process in BP. Cell periphery, non-membrane-bound organelle, intracellular non-membrane-bound organelle, cell wall, and external encapsulating structure were mainly enriched in CC. Coenzyme binding, DNA-binding transcription factor activity, heme binding, tetrapyrrole binding, iron ion binding, transferase activity, and transferring glycosyl groups were mainly enriched in MF.

#### 3.5.3. GO Enrichment Analysis Diagram of DEGs in Each Root Comparison Group

As shown in Figure 7G, the DEGs in the R200 vs. R0 comparison group were mainly enriched in response to stress, ion transport, lipid metabolic process, cell communication, cellular response to stimulus, organic acid metabolic process, and oxoacid metabolic process in BP. Cell periphery, non-membrane-bound organelle, intracellular non-membrane-bound organelle, cell wall, external encapsulating structure, and ribonucleoprotein complex were mainly enriched in CC. Heme binding, tetrapyrrole binding, transferase activity, transferring glycosyl groups, iron ion binding and oxidoreductase activity, and acting on paired donors, with incorporation or reduction of molecular oxygen, were mainly enriched in MF.

As shown in Figure 7H, the DEGs in the R400 vs. R0 comparison group were mainly enriched in ion transport, response to stress, cell communication, lipid metabolic process, cellular response to stimulus, cation transport, signal transduction, and signaling in BP. Cell periphery, non-membrane-bounded organelle, intracellular non-membrane-bounded organelle, cell wall, and external encapsulating structure were mainly enriched in CC. Transferase activity, transferring glycosyl groups, heme binding, tetrapyrrole binding, hydrolase activity, acting on glycosyl bonds, transferase activity, transferring hexosyl groups and hydrolase activity, and hydrolyzing O-glycosyl compounds were mainly enriched in MF.

As shown in Figure 7I, the DEGs in the R400 vs. R200 comparison group were mainly enriched in cellular amide metabolic process, cellular response to stimulus, amide biosynthetic process, cell communication, ion transport, peptide metabolic process, peptide biosynthetic process, and translation in BP. Cell periphery, external encapsulating structure, cell wall, non-membrane-bound organelle, and intracellular non-membrane-bound organelle were mainly enriched in CC. Hydrolase activity, acting on glycosyl bonds, hydrolase activity, hydrolyzing O-glycosyl compounds, transferase activity, transferring glycosyl groups, nucleoside-triphosphatase activity, pyrophosphatase activity, hydrolase activity, acting on acid anhydrides, in phosphorus-containing anhydrides, hydrolase activity, and acting on acid anhydrides were mainly enriched in MF.

Thus, the BPs were mainly concentrated in cell communication, ion transport, cellular response to stimulus, signaling and signal transmission, lipid metabolic process response to stress, and organic acid metabolic process. Among the MFs, the most annotated subclasses of DEGs were transferase activity, transferring glycosyl groups, DNA binding, transcription factor activity, hydrolase activity, acting on glycosyl bonds, transcription factor activity, transferring hexyl groups, acting on acid anhydrides, and hydrolyzing O-glycosyl compounds. Among the major categories of CC, the most annotated sub-categories of DEGs were organelle part, intracellular organelle part, cell perimeter, membrane protein complex, intracellular non-membrane bound organelle, non-membrane-bound organelle, cell periphery, cell wall, and external encapsulating structure

### 3.6. KEGG Pathway Enrichment Analysis of DEGs

The KEGG database (http://www.genome.jp/kegg/ (accessed on 6 October 2021)) provides knowledge about genomes and their relationships to biological systems, such as cells and whole organisms, as well as their interactions with the environment [59]. To further analyze the differences in gene expression functions in response to different concentrations of salt stress by *P. talassica* × *P. euphratica*, stem, root, and leaf samples of DEGs in each comparison group were subjected to a KEGG pathway enrichment analysis. From the KEGG enrichment results, the 20 most significant KEGG pathways were selected, and scatter diagrams were constructed. If there were less than 20 pathways, all the pathways were included. In Figure 8A–I, the abscissa represents the ratio of the number of differentially annotated genes to the total number of differentially annotated genes in the KEGG pathway, and the ordinate represents the KEGG pathway. The size of each dot represents the number of genes annotated to the KEGG pathway, and the color, from red to purple, represents the significance of enrichment.

#### 3.6.1. KEGG Pathway Enrichment Analysis Diagram of DEGs in Each Leaf Comparison Group

To further analyze the differential gene expression functions, a KEGG pathway enrichment analysis was carried out on the DEGs obtained from the three leaf comparison groups—L200 vs. L0, L400 vs. L0 and L400 vs. L200—of *P. talassica* × *P. euphratica*. The enrichment results are shown in Figure 8A–C. 

As shown in Figure 8A, in the L200 vs. L0 comparison group, the largest number of differential genes annotated to the pathway was ‘protein processing in endoplasmic reticulum’, and the number of DEGs was 60. The second most enriched pathways were ‘plant–pathogen interaction’ and ‘carbon metabolism’, with 37 DEGs each. There were 32 DEGs in both ‘glutathione metabolism’ and ‘biosynthesis of amino acids’. As shown in Figure 8B, in the L400 vs. L0 comparison group, the largest number of DEGs was annotated to the pathway of ‘plant hormone signal transmission’, having 72 DEGs. This was followed by ‘protein processing in endoplasmic reticulum’ and ‘carbon metabolism’, each with 65 DEGs. There were 57 DEGs in ‘biosynthesis of amino acids’, and there were 53 DEGs in ‘plant–pathogen interaction’. As shown in Figure 8C, in the L400 vs. L200 comparison group, the largest number of DEGs was annotated to the pathway of ‘biosynthesis of amino acids’, having 80 DEGs. The second most annotated pathway was ‘carbon metabolism’, with 74 DEGs. There were 62 DEGs in ‘plant hormone signal transmission’, and there were 51 DEGs in ‘cystaine and methionine metabolism’.

Thus, candidate genes involved in the salt stress response of *P. talassica* × *P. euphratica* may be screened from the metabolic pathways of ‘protein processing in endoplasmic reticulum’, ‘plant–pathogen interaction’, ‘carbon metabolism’, ‘glutathione metabolism’, ‘biosynthesis of amino acids’, ‘plant hormone signal transduction’, and ‘protein processing in endoplasmic reticulum’.

#### 3.6.2. KEGG Pathway Enrichment Analysis Diagram of DEGs in Each Comparison Group of Stem

KEGG pathway enrichment analysis was carried out on the DEGs obtained from the three stem comparison groups S200 vs. S0, S400 vs. S0, and S400 vs. S200 of *P. talassica* × *P. euphratica*. The enrichment results are shown in Figure 8D–F. As shown in Figure 8D, in the S200 vs. S0 comparison group, the largest number of differential genes was annotated to the pathway of ‘carbon metabolism’, and the number of DEGs was 140. The second most annotated pathway was ‘biosynthesis of amino acids’, with 115 DEGs. There were 112 DEGs in ‘plant–pathogen interaction’. There were 109 DEGs in ‘plant hormone signal transmission’. As shown in Figure 8E, in the S400 vs. S0 comparison group, the largest number of DEGs was annotated to the pathway of ‘carbon metabolism’, and the number of DEGs was 149. The second most annotated pathways were ‘plant hormone signal transduction’ and ‘biosynthesis of amino acids’, each with 117 DEGs. There were 108 DEGs in ‘plant–pathogen interaction’. As shown in Figure 8F, in the S400 vs. S200 comparison group, the largest number of DEGs was annotated to the pathway of ‘carbon metabolism’, with 42 DEGs. The second most annotated pathway was ‘glycolysis/gluconeogenesis’, with 41 DEGs. There were 39 DEGs in ‘biosynthesis of amino acids’. There were 35 DEGs in ‘plant hormone signal transmission’. Thus, candidate genes for salt stress responses of *P. talassica* × *P. euphratica* may be screened from the metabolic pathways of ‘carbon metabolism’, ‘biosynthesis of amino acids’, ‘plant–pathogen interaction’, ‘plant hormone signal transduction’, and ‘glycolysis/gluconeogenesis’.

#### 3.6.3. KEGG Pathway Enrichment Analysis Diagram of DEGs in Each Root Comparison Group

KEGG pathway enrichment analysis was carried out on the DEGs obtained from three root comparison groups: R200 vs. R0, R400 vs. R0, and R400 vs. R200 of *P. talassica* × *P. euphratica*. The enrichment results are shown in Figure 8G–I.

As shown in Figure 8G, in the R200 vs. R0 comparison group, the largest number of 85 DEGs was annotated to the ‘plant hormone signal transmission pathway’. The second most annotated pathway was ‘carbon metabolism’, with 56 DEGs. There were 52 DEGs in ‘biosynthesis of amino acids’. There were 51 DEGs in ‘MAPK-signaling pathway–plant’. As shown in Figure 8H, in the R400 vs. R0 comparison group, the largest number of DEGs was annotated to the pathway of ‘plant hormone signal transmission’, with 87 DEGs. The second most annotated pathway was ‘phenylpropanoid biosynthesis’, with 64 DEGs. There were 61 DEGs in ‘plant–pathogen interaction’. There were 60 DEGs in ‘MAPK-signaling pathway–plant’. As shown in Figure 8I, in the R400 vs. R200 comparison group, the largest number of DEGs was annotated to the pathway of ‘plant hormone signal transmission’, with 36 DEGs. The second was ‘plant–pathogen interaction’, with 33 DEGs. There were 32 DEGs in ‘phenylpropanoid biosynthesis’. There were 25 DEGs in ‘carbon metabolism’. There were 24 DEGs in ‘biosynthesis of amino acids’.

Thus, candidate genes for salt-stress response of *P. talassica* × *P. euphratica* may be screened from the metabolic pathways of ‘plant hormone signal transduction’, ‘carbon metabolism’, ‘biosynthesis of amino acids’, ‘MAPK-signaling pathway–plant’, ‘plant–pathogen interaction’, and ‘phenylpropanoid biosynthesis’.

### 3.7. Mutation Loci Analysis

Using *P. euphratica* as the reference genome, after GATK was used for mutation loci detection, a statistical analysis was performed for mutation locus in accordance with the SnpEff (Download and install—SnpEff & SnpSift Documentation (pcingola.github.io)) annotation information, as shown in Figure 9A–C This mainly includes functional, regional, and influential statistics of variation loci. SNP function was statistically analyzed from three aspects, synonymous, missense, and nonsense mutations. The SNP function statistical analysis results of each sample are shown in Figure 9A. Statistical analysis and mapping of SNP impact were carried out at four levels: HIGH, MODERATE, LOW, and MODIFIER (no phenotypic effects on their own, only when they co-exist with other mutation sites). The SNP impact statistics results of each sample are shown in Figure 9B. Statistical analysis and mapping of SNP regions were conducted for EXON, INTRON, INTERGENIC, and other gene structures, and the SNP region statistics results of each sample are shown in Figure 9C.

## 4. Discussion

Studies using transcriptome sequencing to analyze salt tolerance in plants have been increasing in the past few years [58,59,60,61,62,63,64]. In this study, the salt tolerance of *P. talassica × P. euphratica* under different salt concentrations was comprehensively analyzed by using RNA-seq. 

Results revealed that the DEGs in roots, stems, and leaves had distinct expression patterns. The following findings hold valid: the total number of DEGs in stems > the total number of DEGs in leaves > the total number of DEGs in roots. The reason is because under salt stress, DEGs with distinct expression patterns amongst roots, stems, and leaves were ideal targets for further functional research to better understand the more specific molecular mechanisms of salt tolerance [62].

By screening the expression levels of DEGs, the possible salt tolerance genes in the roots, stems, and leaves of *P. talassica × P. euphratica* were further explored. The highly expressed DEGs in each comparison group of leaves such as *LOC105129452*, *LOC105129935*, and *LOC105114274* belong to the AP2/ERF transcription factor family. *LOC105136804*, *LOC105124563*, and *LOC105124001* belong to the NAC (NAM,ATAF1,2,CUC) transcription factor family. *LOC105131213*, *LOC105116811*, and *LOC105132264* belong to the WRKY transcription factor family. *LOC105127881*, *LOC105142149*, *LOC105115176*, *LOC105141220*, *LOC105128502*, and *LOC105131234* belong to the bZIP transcription factor family. The highly expressed DEGs in each comparison group of stems such as *LOC105119414*, *LOC105129607*, and *LOC105122741* belong to the AP2/ERF transcription factor family. *LOC105142033*, *LOC105112164*, and *LOC105116962* belong to the NAC (NAM,ATAF1,2,CUC) transcription factor family. *LOC105119522*, *LOC105131256*, and *LOC105115491* belong to the WRKY transcription factor family. *LOC105141088*, *LOC105141220*, *LOC105137881*, *LOC105126640*, *LOC105134677*, and *LOC105134493* belong to the bZIP transcription factor family. The highly expressed DEGs in each comparison group of roots such as *LOC105108844*, *LOC105107217*, and *LOC105120059* belong to the AP2/ERF transcription factor family. *LOC105135115*, *LOC105127231*, and *LOC105116469* belong to the NAC (NAM,ATAF1,2,CUC) transcription factor family. *LOC105114369*, *LOC105124330*, and *LOC105127348* belong to the WRKY transcription factor family. *LOC105134637*, *LOC105115176*, *LOC105115119*, *LOC105128502*, *LOC105112947*, and *LOC105124801* belong to the bZIP transcription factor family.

Plants often activate an array of defense responses in response to abiotic and biotic stresses, including inducible expression of a set of stress-related genes that are regulated directly or indirectly by various types of transcription factors [65]. Several classes of transcription factors, including AP2/ERF, bZIP, NAC (NAM, ATAF, and CUC), and WRKY, are reported to be associated with plant defense [66,67,68,69]. NAC (NAM, ATAF, and CUC) domain proteins are plant-specific transcriptional factors that play diverse roles in plant growth and development [70,71]. NAC (NAM, ATAF, and CUC) transcription factors are also involved in plant responses to biotic and abiotic stress processes, including high salt [72], drought [73], freezing [74], and viral infection [75]. NAC (NAM, ATAF, and CUC) plant transcription factors regulate essential processes in development, stress responses, and nutrient distribution in important crop and model plants (rice, *Populus*, *Arabidopsis*), making them crucial in crop improvement and production [76].

The AP2/ERF transcription factors are known to regulate diverse processes of plant development and stress responses. In addition, AP2/ERF transcription factors are ideal candidates for crop growth, development, and improvement because their overexpression enhances tolerances to drought and salt stress in the transgenic plants [77]. Various AP2/ERF transcription factors have been successfully identified and investigated in some plants, including *Arabidopsis*, rice [78,79], poplar (*P. tricocarpa*) [80], and wheat (*Triticum aestivum*) [81]. WRKY transcription factors, a family of regulatory genes, were first identified in plants [82,83,84]. Numerous WRKY transcription factors are key regulators of many plant processes, including the responses to biotic and abiotic stresses, senescence, seed dormancy, and seed germination [85]. A large number of WRKY transcription factors have been reported from *Arabidopsis*, rice, and other higher plants [86]. The *Populus* genome contains at least 100 WRKY genes [87]. Increasing evidence has confirmed the importance of WRKY transcription factors in poplar defense processes, and several poplar WRKY genes have been identified to be involved in defense response [88].

The bZIP transcription factor gene family is one of the largest and most diverse families in plants. The bZIP proteins regulate numerous growth and developmental processes [89]. They are also involved in responses to various abiotic/biotic stimuli, such as drought [90,91] and high salinity [77]. Some of these bZIP factors confer disease resistance [92] and trigger expression of defense-related genes in systemic acquired resistance (SAR) [93]. bZIP-type transcriptional factors are involved in developmental and physiological processes in response to stresses and are important for various plants to withstand adverse environmental conditions [94,95].

Stress-resistant transcription factors, such as AP2/ERF, NAC (NAM, ATA, and CUC), WRKY, and bZIP, are all involved in the detection of early salt stress in poplars and other plants. The analysis found that these differential genes with high expression in the roots, stems, and leaves of *P. talassica × P. euphratica* also existed in its male parent *P. euphratica*. These genes are likely to be the key salt-tolerant genes that exist in *P. talassica × P. euphratica* to resist external salt stress. In the future, the functions of these genes in *P. talassica × P. euphratica* and other tree species should be further analyzed to provide some reference for the planting of *P. talassica × P. euphratica* in saline-alkali land.

GO enrichment analysis showed that many DEGs were related to ‘material metabolism’, ‘signal transmission’, and other processes, which was consistent with the annotation results of *Reaumuria trigyna* [96]. The genes involved in the above processes, such as ‘signal transmission’, ‘ion transport’, ‘cell communication’, ‘cell periphery’, and ‘cell wall’, may be involved in *P. talassica × P. euphratica* responses to salt stress. Salt stress disturbs the normal growth and development of plants. Modifications of the cell wall are common defense responses when plants are subjected to abiotic and biotic stresses. In our study, most of the genes related to cell wall and growth were upregulated under salt stress conditions [97]. KEGG enrichment analysis revealed that DEGs were mainly enriched in ‘plant hormone signal transduction’ and ‘MAPK-signaling pathway–plant’, which were highly annotated, may be involved in *P. talassica × P. euphratica* responses to salt stress. This is similar to the research results of Zhan Jiang Han et al. [58]. 

For plants under abiotic or biotic stress, signal transduction is very crucial for adjustment in such unfavorable conditions [98,99,100,101]. Plants respond to abiotic stresses by regulating complex signaling networks, helping plants adapt to stress and thus enhancing their growth and development [102]. Furthermore, hormone signal transduction is involved in plant growth regulation and root development under salt stress conditions [103]. KEGG-enriched pathways of these DEGs were similar, and *P. talassica × P. euphratica* may have a special response mechanism to salt stress.

Through GO functional enrichment analysis and KEGG enrichment analysis, the metabolic pathways and molecular functions of the DEGs enriched in *P. talassica × P. euphratica* were clarified, and the response mechanism of salt stress in *P. talassica × P. euphratica* was revealed at the molecular level. A unique salt tolerance mechanism may provide a reference for further mining of salt tolerance genes of *P. talassica × P. euphratica*. Subsequent transgenic, proteomics, and metabolomics studies should be carried out to explain the unique salt tolerance mechanism of *P. talassica × P. euphratica* at multiple levels.

## 5. Conclusions

In this study, the stems, roots, and leaves of *P. talassica* × *P. euphratica* seedlings treated with different concentrations of NaCl solution were subjected to transcriptome sequencing. A total of 1,109,438,718 clean reads, accounting for 97.08% of the raw reads, were obtained. The raw reads of each group contained over 94% high-quality reads, producing 166.41 G of high-quality reads. The obtained data were screened under the conditions of |log_2_ (fold change)| ≥ 1 and padj ≤ 0.05 to obtain DEGs, which illustrated the differences among salt treatments (200 and 400 mmol/L NaCl) and controls. A total of 41,617 DEGs were obtained in each comparison group. Using |log2 (fold change)| ≥ 5 as a selection criteria, a total of 45 DEGs in AP2/ERF, NAC (NAM, ATAF, and CUC), WRKY, and bZIP transcription factor families in roots, stems, and leaves of *P. talassica × P. euphratica* were screened in this study. *P. talassica* × *P. euphratica* have up- and downregulated genes. Generally, the number of upregulated genes in *P. talassica* × *P. euphratica* was greater than the number of downregulated genes. Thus, it may be inferred that the effects of salt stress on gene expression trends is mainly upregulated. GO functional enrichment analysis and KEGG pathway enrichment analysis further clarified the metabolic pathways and molecular functions of the DEGs enriched in *P. talassica* × *P. euphratica.* This finding can contribute to the screening salt-stress-responsive candidate genes of *P. talassica* × *P. euphratica* from the more enriched metabolic pathways in the future. Through further analyses, a theoretical basis for the cultivation and planting of stress-resistant *P. talassica* × *P. euphratica* varieties will be established.

## Figures and Tables

**Figure 1 genes-13-01032-f001:**
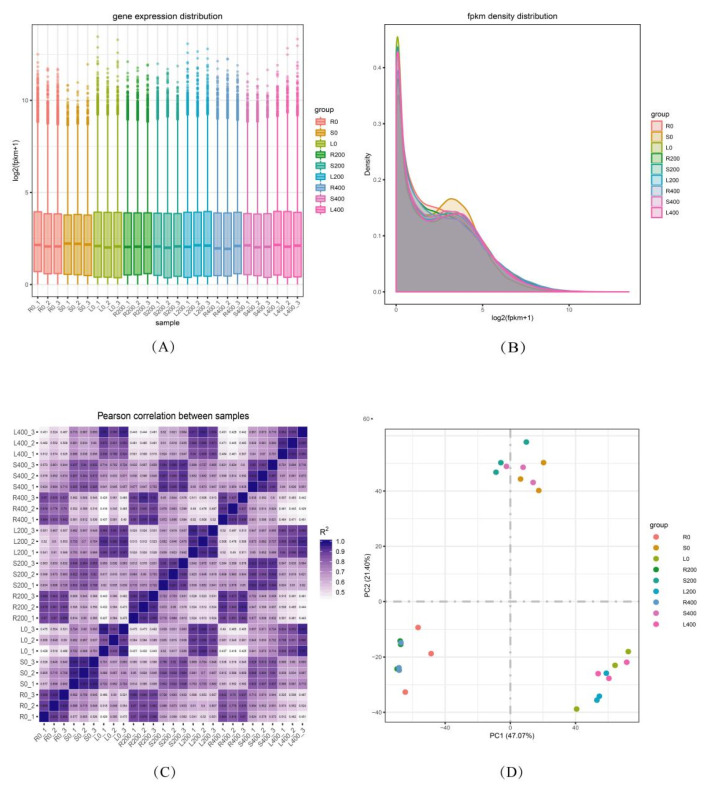
Quantitative analysis of gene expression. (**A**) Distribution box plot of gene expression levels in different samples. (**B**) Density distribution diagram of FPKM values. (**C**) Correlation heat map. (**D**) PCA diagram of samples.

**Figure 2 genes-13-01032-f002:**
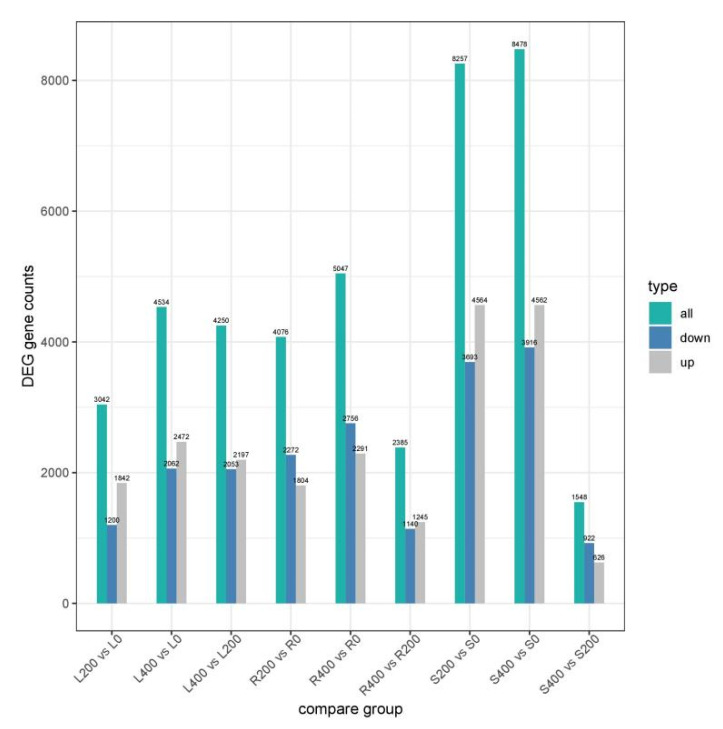
Differentially expressed genes of *P. talassica × P. euphratica* in different comparison groups.

**Figure 3 genes-13-01032-f003:**
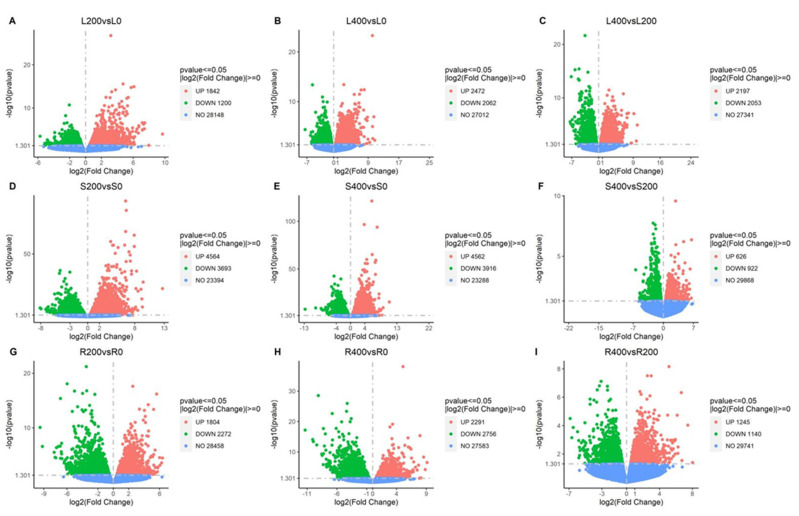
Volcano diagram of differential gene expression levels in each comparison group. (**A**) L200 vs. L0; (**B**) L400 vs. L0; (**C**) L400 vs. L200; (**D**) S200 vs. S0; (**E**) S400 vs. S0; (**F**) S400 vs. S200; (**G**) R200 vs. R0; (**H**) R400 vs. R0; (**I**) R400 vs. R200.

**Figure 4 genes-13-01032-f004:**
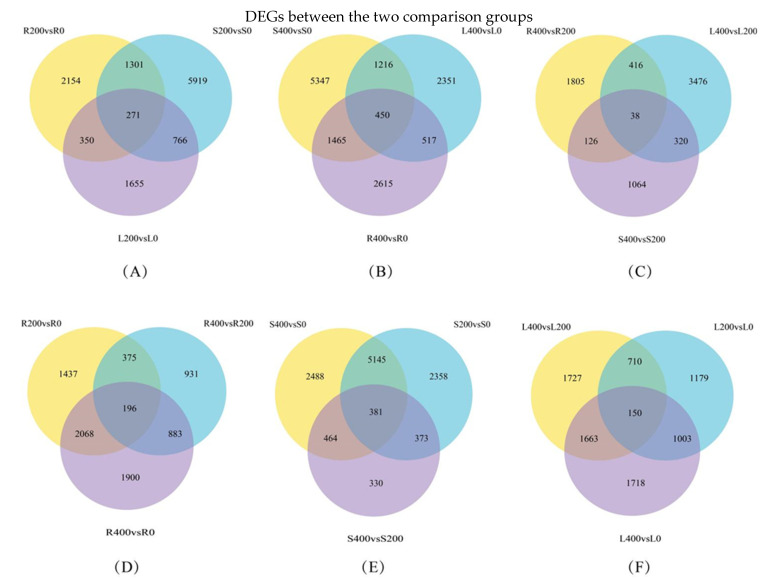
Venn diagram of differential gene expression levels in each comparison group. (**A**) R200 vs. R0_ S200 vs. S0_L200 vs. L0 group; (**B**) S400 vs. S0_L400 vs. L0_R400 vs. R0 group; (**C**) R400 vs. R200_L400 vs. L200_ S400 vs. S200 group; (**D**) R200 vs. R0_R400 vs. R200_R400 vs. R0 group; (**E**) S400 vs. S0_S200 vs. S0_S400 vs. S200 group; (**F**) L200 vs. L0_L400 vs. L0_L400 vs. L200 group. Note: As shown in (**A**–**F**), the number of genes expressed in each comparison group and their overlapping relationships are shown. The sum of the numbers in each circle represents the total number of DEGs in the comparison combination, and the overlapping part of the circle represents the common.

**Figure 5 genes-13-01032-f005:**
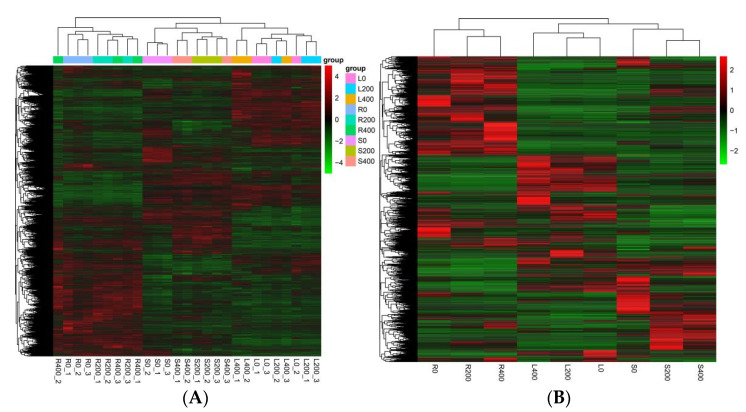
Cluster heat maps of differentially expressed genes. Clustering between (**A**) samples and (**B**) groups.

**Figure 6 genes-13-01032-f006:**
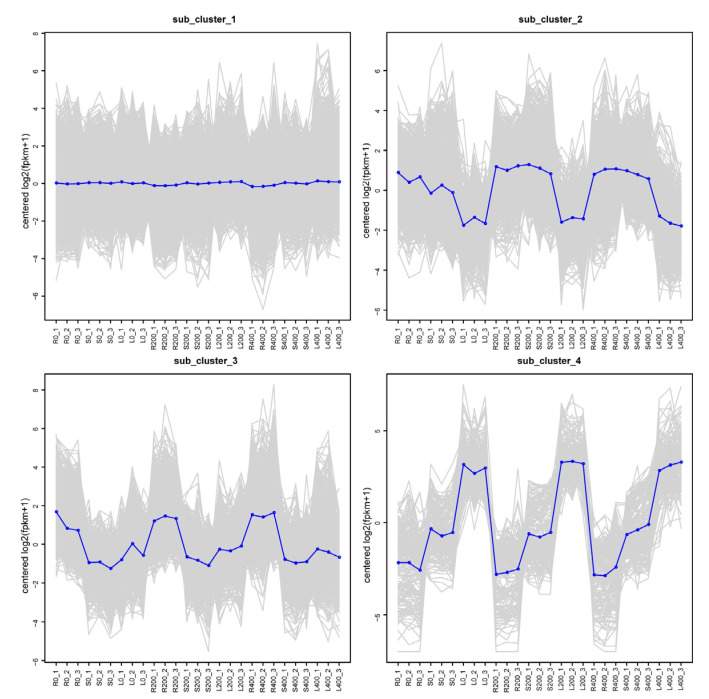
Trend analysis.

**Figure 7 genes-13-01032-f007:**
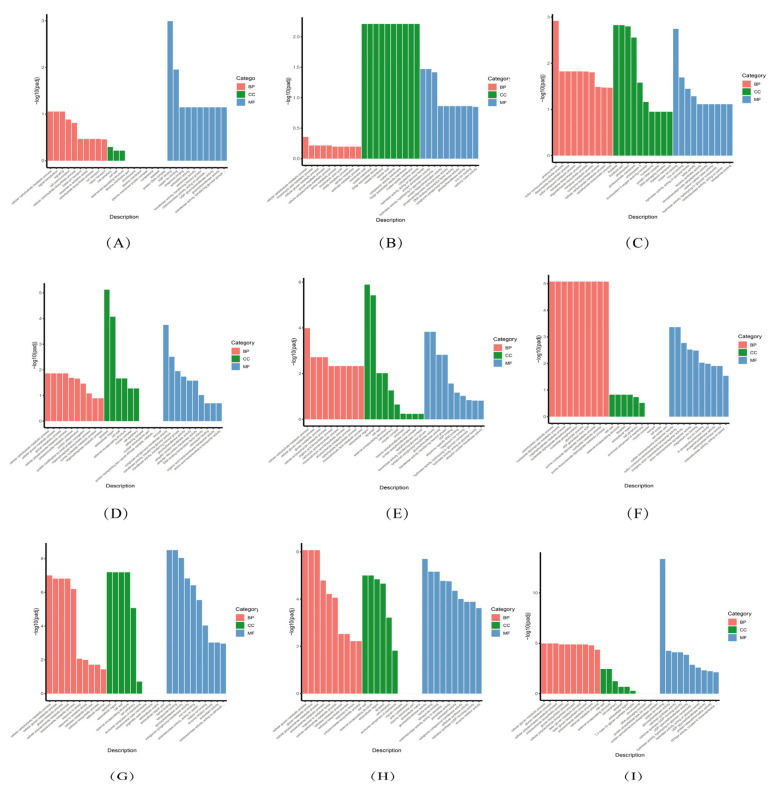
GO enrichment analysis of differentially expressed genes in each comparison group of leaf, stem and root. (**A**) L200 vs. L0; (**B**) L400 vs. L0; (**C**) L400 vs. L200; (**D**) S200 vs. S0; (**E**) S400 vs. S0; (**F**) S400 vs. S200; (**G**) R200 vs. R0; (**H**) R400 vs. R0; (**I**) R400 vs. R200.

**Figure 8 genes-13-01032-f008:**
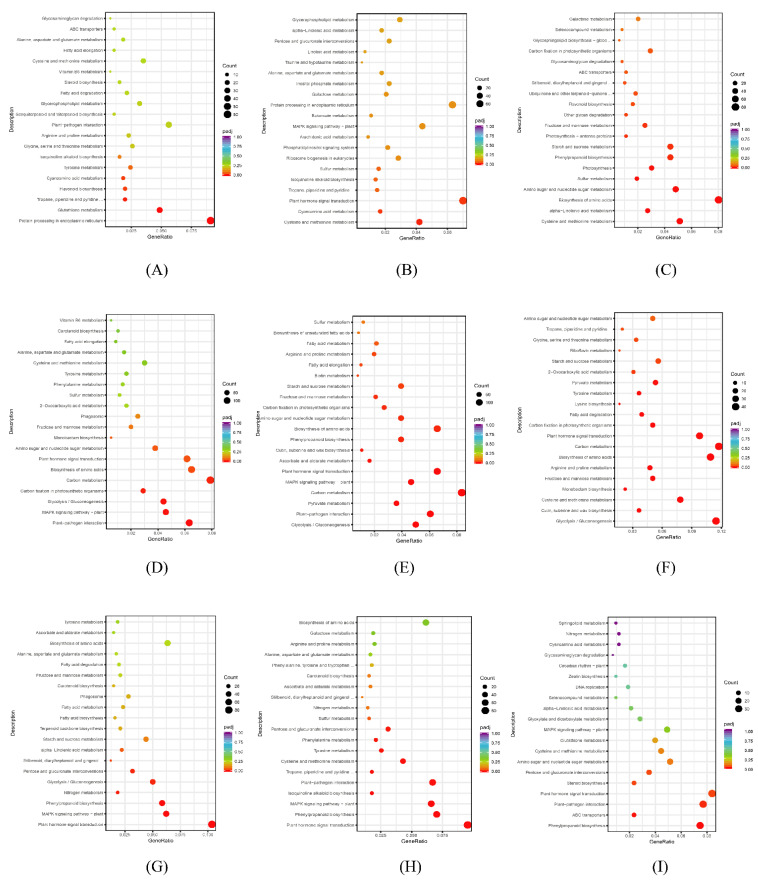
KEGG pathway enrichment analysis of differentially expressed genes in each comparison group of leaf, stem and root. (**A**) L200 vs. L0; (**B**) L400 vs. L0; (**C**) L400 vs. L200; (**D**) S200 vs. S0; (**E**) S400 vs. S0; (**F**) S400 vs. S200; (**G**) R200 vs. R0; (**H**) R400 vs. R0; (**I**) R400 vs. R200.

**Figure 9 genes-13-01032-f009:**
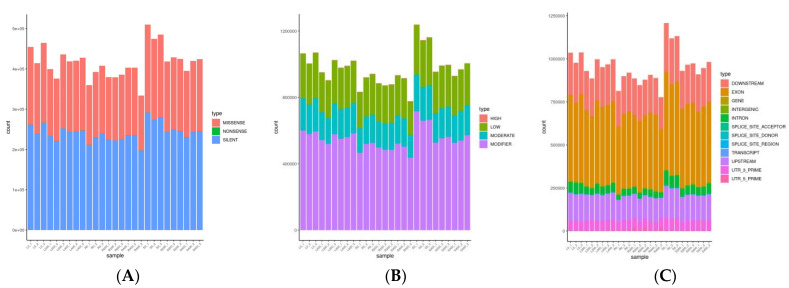
Mutation loci analysis. (**A**) Functional statistics of mutants. (**B**) Statistical diagram of influences of locus variation. (**C**) Regional statistical map of variation loci.

**Table 1 genes-13-01032-t001:** Information of RNA-seq data.

Sample	Library	Raw_Reads	Clean_Reads	Clean_Bases	Error_Rate	Q20	Q30	GC_Pct
L0_1	FRAS210141894-1r	43,229,878	41,916,070	6.29 G	0.03	97.72	93.28	43.80
L0_2	FRAS210141895-1r	43,503,400	41,412,972	6.21 G	0.03	97.72	93.30	42.92
L0_3	FRAS210141896-1r	41,700,782	40,131,530	6.02 G	0.03	97.64	93.04	43.93
L200_1	FRAS210141903-1r	42,337,890	41,376,760	6.21 G	0.03	97.75	93.37	44.00
L200_2	FRAS210141904-1r	41,637,272	40,851,878	6.13 G	0.03	97.72	93.30	43.90
L200_3	FRAS210141905-1r	41,808,814	40,466,298	6.07 G	0.03	97.77	93.38	43.56
L400_1	FRAS210141912-1r	43,477,378	41,505,660	6.23 G	0.03	97.72	93.27	43.25
L400_2	FRAS210141913-1r	40,334,862	39,717,696	5.96 G	0.03	97.58	92.95	43.92
L400_3	FRAS210141914-1r	43,098,376	42,022,164	6.30 G	0.03	97.71	93.32	43.79
R0_1	FRAS210141888-1r	41,219,840	38,942,918	5.84 G	0.03	97.31	92.50	44.26
R0_2	FRAS210141889-1r	40,474,488	39,722,168	5.96 G	0.03	97.66	93.17	43.25
R0_3	FRAS210141890-1r	41,990,470	41,326,130	6.20 G	0.03	97.70	93.25	43.58
R200_1	FRAS210141897-1r	41,346,852	40,288,012	6.04 G	0.03	97.60	93.07	43.87
R200_2	FRAS210141898-1r	41,922,834	40,528,650	6.08 G	0.03	97.59	93.08	44.29
R200_3	FRAS210141899-1r	42,625,060	41,322,596	6.20 G	0.03	97.54	93.02	44.55
R400_1	FRAS210141906-2r	47,618,458	46,825,304	7.02 G	0.03	97.39	92.57	44.77
R400_2	FRAS210141907-2r	44,838,230	43,672,332	6.55 G	0.03	97.40	92.61	44.64
R400_3	FRAS210141908-1r	40,153,594	38,188,658	5.73 G	0.03	97.58	93.05	44.60
S0_1	FRAS210141891-1r	42,998,830	41,953,190	6.29 G	0.03	97.68	93.21	43.01
S0_2	FRAS210141892-1r	41,626,198	40,928,406	6.14 G	0.03	97.67	93.14	42.92
S0_3	FRAS210141893-1r	42,162,498	40,897,322	6.13 G	0.03	97.75	93.32	43.04
S200_1	FRAS210141900-1r	41,365,530	39,960,594	5.99 G	0.03	97.58	93.11	43.65
S200_2	FRAS210141901-1r	41,693,852	40,826,972	6.12 G	0.03	97.50	92.84	43.60
S200_3	FRAS210141902-1r	43,498,006	42,704,462	6.41 G	0.03	97.74	93.33	43.13
S400_1	FRAS210141909-1r	43,083,076	41,548,528	6.23 G	0.03	97.65	93.17	43.24
S400_2	FRAS210141910-1r	42,854,862	41,521,356	6.23 G	0.03	97.70	93.27	43.29
S400_3	FRAS210141911-1r	40,153,198	38,880,092	5.83 G	0.03	97.44	92.77	42.81

**Table 2 genes-13-01032-t002:** Statistics of the comparison between samples and reference genomes.

Sample	Total_Reads	Total_Map	Unique_Map	Multi_Map	Read1_Map	Read2_Map	Positive_Map	Negative_Map	Splice_Map	Unsplice_Map	Proper_Map
L0_1	41,916,070	34,099,861	31,929,576	2,170,285	16,031,977	15,897,599	15,964,215	15,965,361	14,181,366	17,748,210	28,988,606
		(81.35%)	(76.18%)	(5.18%)	(38.25%)	(37.93%)	(38.09%)	(38.09%)	(33.83%)	(42.34%)	(69.16%)
L0_2	41,412,972	32,664,958	30,703,326	1,961,632	15,406,956	15,296,370	15,331,819	15,371,507	11,989,541	18,713,785	27,601,396
		(78.88%)	(74.14%)	(4.74%)	(37.20%)	(36.94%)	(37.02%)	(37.12%)	(28.95%)	(45.19%)	(66.65%)
L0_3	40,131,530	32,731,445	30,693,348	2,038,097	15,420,588	15,272,760	15,336,379	15,356,969	13,627,713	17,065,635	27,583,972
		(81.56%)	(76.48%)	(5.08%)	(38.43%)	(38.06%)	(38.22%)	(38.27%)	(33.96%)	(42.52%)	(68.73%)
L200_1	41,376,760	32,898,394	30,611,563	2,286,831	15,356,087	15,255,476	15,298,246	15,313,317	12,888,498	17,723,065	27,751,170
		(79.51%)	(73.98%)	(5.53%)	(37.11%)	(36.87%)	(36.97%)	(37.01%)	(31.15%)	(42.83%)	(67.07%)
L200_2	40,851,878	32,073,201	29,883,288	2,189,913	14,986,813	14,896,475	14,936,879	14,946,409	12,214,282	17,669,006	27,036,076
		(78.51%)	(73.15%)	(5.36%)	(36.69%)	(36.46%)	(36.56%)	(36.59%)	(29.90%)	(43.25%)	(66.18%)
L200_3	40,466,298	32,754,785	30,638,096	2,116,689	15,357,668	15,280,428	15,310,291	15,327,805	12,668,641	17,969,455	27,793,958
		(80.94%)	(75.71%)	(5.23%)	(37.95%)	(37.76%)	(37.83%)	(37.88%)	(31.31%)	(44.41%)	(68.68%)
L400_1	41,505,660	33,238,571	31,112,328	2,126,243	15,607,894	15,504,434	15,538,117	15,574,211	12,673,945	18,438,383	28,325,300
		(80.08%)	(74.96%)	(5.12%)	(37.60%)	(37.35%)	(37.44%)	(37.52%)	(30.54%)	(44.42%)	(68.24%)
L400_2	39,717,696	31,919,467	29,885,359	2,034,108	15,027,743	14,857,616	14,941,370	14,943,989	12,497,015	17,388,344	26,970,556
		(80.37%)	(75.24%)	(5.12%)	(37.84%)	(37.41%)	(37.62%)	(37.63%)	(31.46%)	(43.78%)	(67.91%)
L400_3	42,022,164	33,553,507	31,383,445	2,170,062	15,742,857	15,640,588	15,688,348	15,695,097	13,645,448	17,737,997	28,539,948
		(79.85%)	(74.68%)	(5.16%)	(37.46%)	(37.22%)	(37.33%)	(37.35%)	(32.47%)	(42.21%)	(67.92%)
R0_1	38,942,918	29,175,374	27,383,231	1,792,143	13,782,210	13,601,021	13,664,279	13,718,952	9,546,470	17,836,761	24,479,086
		(74.92%)	(70.32%)	(4.60%)	(35.39%)	(34.93%)	(35.09%)	(35.23%)	(24.51%)	(45.80%)	(62.86%)
R0_2	39,722,168	30,801,737	28,995,952	1,805,785	14,556,757	14,439,195	14,473,119	14,522,833	10,181,574	18,814,378	26,065,798
		(77.54%)	(73.00%)	(4.55%)	(36.65%)	(36.35%)	(36.44%)	(36.56%)	(25.63%)	(47.36%)	(65.62%)
R0_3	41,326,130	32,315,221	30,390,025	1,925,196	15,250,293	15,139,732	15,172,739	15,217,286	11,044,773	19,345,252	27,505,356
		(78.20%)	(73.54%)	(4.66%)	(36.90%)	(36.63%)	(36.71%)	(36.82%)	(26.73%)	(46.81%)	(66.56%)
R200_1	40,288,012	31,451,369	29,490,159	1,961,210	14,795,186	14,694,973	14,720,654	14,769,505	11,114,536	18,375,623	26,546,638
		(78.07%)	(73.20%)	(4.87%)	(36.72%)	(36.47%)	(36.54%)	(36.66%)	(27.59%)	(45.61%)	(65.89%)
R200_2	40,528,650	31,057,737	29,136,961	1,920,776	14,612,559	14,524,402	14,547,707	14,589,254	10,636,348	18,500,613	26,202,788
		(76.63%)	(71.89%)	(4.74%)	(36.05%)	(35.84%)	(35.89%)	(36.00%)	(26.24%)	(45.65%)	(64.65%)
R200_3	41,322,596	30,250,140	28,378,130	1,872,010	14,241,200	14,136,930	14,165,430	14,212,700	10,488,271	17,889,859	25,446,492
		(73.20%)	(68.67%)	(4.53%)	(34.46%)	(34.21%)	(34.28%)	(34.39%)	(25.38%)	(43.29%)	(61.58%)
R400_1	46,825,304	34,091,693	31,999,767	2,091,926	16,001,871	15,997,896	15,983,812	16,015,955	11,468,760	20,531,007	28,565,350
		(72.81%)	(68.34%)	(4.47%)	(34.17%)	(34.17%)	(34.13%)	(34.20%)	(24.49%)	(43.85%)	(61.00%)
R400_2	43,672,332	34,574,074	32,289,809	2,284,265	16,166,727	16,123,082	16,116,947	16,172,862	11,606,074	20,683,735	29,067,328
		(79.17%)	(73.94%)	(5.23%)	(37.02%)	(36.92%)	(36.90%)	(37.03%)	(26.58%)	(47.36%)	(66.56%)
R400_3	38,188,658	25,409,066	23,861,327	1,547,739	11,973,526	11,887,801	11,908,929	11,952,398	8,480,579	15,380,748	21,370,942
		(66.54%)	(62.48%)	(4.05%)	(31.35%)	(31.13%)	(31.18%)	(31.30%)	(22.21%)	(40.28%)	(55.96%)
S0_1	41,953,190	32,267,320	30,360,095	1,907,225	15,242,528	15,117,567	15,157,535	15,202,560	11,350,223	19,009,872	27,252,422
		(76.91%)	(72.37%)	(4.55%)	(36.33%)	(36.03%)	(36.13%)	(36.24%)	(27.05%)	(45.31%)	(64.96%)
S0_2	40,928,406	31,190,534	29,382,790	1,807,744	14,736,558	14,646,232	14,660,622	14,722,168	10,667,910	18,714,880	26,398,080
		(76.21%)	(71.79%)	(4.42%)	(36.01%)	(35.79%)	(35.82%)	(35.97%)	(26.06%)	(45.73%)	(64.50%)
S0_3	40,897,322	31,534,019	29,548,587	1,985,432	14,809,974	14,738,613	14,739,097	14,809,490	11,417,118	18,131,469	26,865,924
		(77.11%)	(72.25%)	(4.85%)	(36.21%)	(36.04%)	(36.04%)	(36.21%)	(27.92%)	(44.33%)	(65.69%)
S200_1	39,960,594	31,613,489	29,619,742	1,993,747	14,855,726	14,764,016	14,775,203	14,844,539	11,488,947	18,130,795	26,757,178
		(79.11%)	(74.12%)	(4.99%)	(37.18%)	(36.95%)	(36.97%)	(37.15%)	(28.75%)	(45.37%)	(66.96%)
S200_2	40,826,972	32,184,287	30,214,916	1,969,371	15,174,062	15,040,854	15,076,017	15,138,899	11,759,858	18,455,058	27,384,426
		(78.83%)	(74.01%)	(4.82%)	(37.17%)	(36.84%)	(36.93%)	(37.08%)	(28.80%)	(45.20%)	(67.07%)
S200_3	42,704,462	32,736,990	30,757,053	1,979,937	15,435,697	15,321,356	15,343,424	15,413,629	12,076,016	18,681,037	27,631,548
		(76.66%)	(72.02%)	(4.64%)	(36.15%)	(35.88%)	(35.93%)	(36.09%)	(28.28%)	(43.74%)	(64.70%)
S400_1	41,548,528	31,715,603	29,766,573	1,949,030	14,916,876	14,849,697	14,844,254	14,922,319	11,238,002	18,528,571	26,809,282
		(76.33%)	(71.64%)	(4.69%)	(35.90%)	(35.74%)	(35.73%)	(35.92%)	(27.05%)	(44.60%)	(64.53%)
S400_2	41,521,356	32,538,126	30,505,240	2,032,886	15,313,023	15,192,217	15,209,470	15,295,770	12,225,058	18,280,182	27,518,922
		(78.36%)	(73.47%)	(4.90%)	(36.88%)	(36.59%)	(36.63%)	(36.84%)	(29.44%)	(44.03%)	(66.28%)
S400_3	38,880,092	30,215,448	28,441,954	1,773,494	14,304,860	14,137,094	14,181,255	14,260,699	10,674,997	17,766,957	25,446,248
		(77.71%)	(73.15%)	(4.56%)	(36.79%)	(36.36%)	(36.47%)	(36.68%)	(27.46%)	(45.70%)	(65.45%)

## Data Availability

The RNA-seq datasets from 27 root, stem, and leaf samples of *P. talassica* × *P. euphratica* were uploaded at the NCBI Sequence Read Archive (SRA) database; PRJNA819946: The hybrid of *P. talassica* and *P. euphratica* Transcriptome. (TaxID: 2929483); Reviewer link: https://submit.ncbi.nlm.nih.gov/subs/ (accessed on 19 May 2022).

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
