# Peer review of "Transcriptomic Profile Analysis of Populus talassica × Populus euphratica Response and Tolerance under Salt Stress Conditions"

_genes, 2022, doi:10.3390/genes13061032_

Round 1
Reviewer 1 Report
Authors have analysed the transcriptome of the root, stem and leaves of a new Populus variety grown under salt stress. Although the topic is interesting, I have the following comments;
The language needs to be improved as many sentences are not complete. For example,
Lines 10-12: To elucidate ..........
Lines 161-162: The basic principle of ............
Although the authors gave very detailed information in the materials and methods and the results sections, the discussion is very weak and needs to be rewritten. For example; in lines 802-808, the authors are repeating the same content in different words. In addition, the first four paragraphs of the discussion are simply repeating the introduction, materials and methods, and results with just one sentence of discussion in lines 840-843. Even this sentence is not supporting any findings.
Author Response
Response to Reviewer 1 Comments
Point 1: The language needs to be improved as many sentences are not complete. For example,
Lines 10-12: To elucidate ..........
Lines 161-162: The basic principle of ............
Response 1: Thank you very much for your valuable comments, I have completed the sentence, and the completed sentence is shown below.
(1)Original sentence: To elucidate the molecular mechanism of salt tolerance and the major differentially expressed genes related to salt tolerance of P. talassica × P. euphratica.
(1)Modified sentence: In order to elucidate the molecular mechanism and find out the major differentially expressed genes of salt tolerance of P. talassica × P. euphratica.
(2)Original sentence: The basic principle of this sequencing technology Sequencing by Synthesis.
(2)Modified sentence: The basic principle of this sequencing technology is sequencing by synthesis.
Point 2: Although the authors gave very detailed information in the materials and methods and the results sections, the discussion is very weak and needs to be rewritten. For example; in lines 802-808, the authors are repeating the same content in different words. In addition, the first four paragraphs of the discussion are simply repeating the introduction, materials and methods, and results with just one sentence of discussion in lines 840-843. Even this sentence is not supporting any findings.
Response 2: Thank you very much for your valuable comments, which are very pertinent. I have made revisions to the discussion section to compare the results of differentially expressed genes that may be related to salt stress response. The specific content of the revision is as follows.
Modified Discussion:
In the past few years, there has been an increasing number of studies using transcriptome sequencing to analyze salt tolerance in plants [58,59,60,61,62,63,64]. In this study, the salt tolerance of P. talassica × P. euphratica under different salt concentrations was comprehensively analyzed by RNA-seq.
Differentially expressed genes analysis results showed that the differentially expressed genes in roots, stems and leaves had different expression patterns. Overall, the following was true: the total number of differentially expressed genes in stems > the total number of differential genes in leaves > the total number of differential genes in roots. This is because differentially expressed genes with different expression patterns among roots, stems and leaves under salt-stress conditions were ideal targets for further functional research to understand the more specific molecular mechanisms of salt tolerance [62].
By screening the expression levels of differentially expressed genes, the possible salt tolerance genes in the roots, stems and leaves of P. talassica × P. euphratica were further explored. The highly expressed differential genes in each comparison group of leaves such as LOC105129452, LOC105129935 and LOC105114274 belong to the AP2/ERF transcription factor family. LOC105136804, LOC105124563 and LOC105124001 belong to the NAC (NAM,ATAF1,2,CUC) transcription factor family. LOC105131213, LOC105116811 and LOC105132264 belong to the WRKY transcription factor family. LOC105127881, LOC105142149, LOC105115176, LOC105141220, LOC105128502 and LOC105131234 belong to the bZIP transcription factor family. The highly expressed differential genes in each comparison group of stems such as LOC105119414, LOC105129607 and LOC105122741 belong to the AP2/ERF transcription factor family. LOC105142033, LOC105112164 and LOC105116962 belong to the NAC (NAM,ATAF1,2,CUC) transcription factor family. LOC105119522 , LOC105131256 and LOC105115491 belong to the WRKY transcription factor family. LOC105141088, LOC105141220, LOC105137881, LOC105126640, LOC105134677 and LOC105134493 belong to the bZIP transcription factor family. The highly expressed differential genes in each comparison group of roots such as LOC105108844, LOC105107217 and LOC105120059 belong to the AP2/ERF transcription factor family. LOC105135115, LOC105127231 and LOC105116469 belong to the NAC (NAM,ATAF1,2,CUC) transcription factor family. LOC105114369, LOC105124330 and LOC105127348 belong to the WRKY transcription factor family. LOC105134637, LOC105115176, LOC105115119, LOC105128502, LOC105112947 and LOC105124801 belong to the bZIP transcription factor family.
In response to abiotic and biotic stresses, plants often activate a battery of defense responses that include inducible expression of a set of stress-related genes, which are regulated directly or indirectly by different types of transcription factors [80]. Several classes of transcription factors, including AP2/ERF, bZIP, NAC (NAM, ATAF, and CUC) and WRKY have been reported to be associated with plant defense [100-103]. NAC (NAM, ATAF, and CUC) domain proteins are plant-specific transcriptional factors known to play diverse roles in plant growth and development [73,74]. NAC (NAM, ATAF, and CUC) transcription factors are also involved in plant responses to biotic and abiotic stress processes, including high salt [75], drought [76], freezing [77], and viral infection [78]. NAC (NAM, ATAF, and CUC) plant transcription factors regulate essential processes in development, stress responses and nutrient distribution in important crop and model plants (rice, Populus, Arabidopsis), which makes them play a huge role in crop improvement and production [79].
The AP2/ERF transcription factors are known to regulate diverse processes of plant development and stress responses. In addition, AP2/ERF transcription factors are ideal candidates for crop growth, development and improvement because their overexpression enhances tolerances to drought, salt stress in the transgenic plants [95]. A variety of AP2/ERF transcription factors have been successfully identified and investigated in some plants, including Arabidopsis, rice [81,82], poplar (Populus tricocarpa) [83], and wheat (Triticum aestivum) [84]. WRKY transcription factors, a family of regulatory genes, were first identified in plants [87–89]. WRKY transcription factors are key regulators of many plant processes, including the responses to biotic and abiotic stresses, senescence, seed dormancy and seed germination [85]. A large number of WRKY transcription factors have been reported from Arabidopsis, rice, and other higher plants [86]. The Populus genome contains at least 100 WRKY genes [91]. Increasing evidence has confirmed important roles of WRKY transcription factors in poplar defense processes, and several poplar WRKY genes have been identified to be involved in defense response [90].
The bZIP transcription factor gene family is one of the largest and most diverse families in plants. Current studies have shown that the bZIP proteins regulate numerous growth and developmental processes [92]. They have also been found to be involved in responses to a variety of abiotic/biotic stimuli, such as drought [93,94], and high salinity [95]. Some of these bZIP factors have been shown to confer disease resistance [96] and to trigger expression of defense-related genes in systemic acquired resistance (SAR) [97].The bZIP-type transcriptional factors are involved in developmental and physiological processes in response to stresses, and are important for various plants to withstand adverse environmental conditions [98,99].
Transcription factor families such as AP2/ERF, NAC (NAM, ATAF, and CUC), WRKY and bZIP are all stress-resistant transcription factors that have been reported to be involved in poplars and other plants sensing of early salt stress. The analysis found that these differential genes with high expression in the roots, stems and leaves of P. talassica × P. euphratica were also existed in its male parent P. euphratica. They are very likely to be the key salt-tolerant genes that exist in P. talassica × P. euphratica to resist external salt stress. In the future, the functions of these genes in P. talassica × P. euphratica and other tree species should be further analyzed, so as to provide some reference for the planting of P. talassica × P. euphratica in saline-alkali land.
A gene ontology functional enrichment analysis showed that, many differentially expressed genes were related to material metabolism, signal transmission and other processes, which was consistent with the annotation results of Reaumuria trigyna [65]. The genes involved in the above processes, such as signal transmission, ion transport, cell communication, cell periphery and cell wall, may be involved in P. talassica × P. euphratica responses to salt stress. Salt stress disturbs the normal growth and development of plants. Modifications of the cell wall are common defense responses when plants are suffering from abiotic and biotic stresses. In our study, most of the genes related to cell wall and growth were up-regulated under salt stress conditions [66]. A Kyoto Encyclopedia of Genes and Genomes pathway enrichment analysis showed that different genes were mainly enriched in Plant hormone signal transduction and MAPK-signaling pathway–plant, which were highly annotated, may be involved in P. talassica × P. euphratica responses to salt stress. This is similar to the research results of Zhan Jiang Han et al [58].
For plants under abiotic or biotic stress, signal transduction is very crucial for adjustment in such unfavorable conditions [67–70]. Plants respond to abiotic stresses by regulating complex signaling networks, which helps plants adapt to stress and thus enhance their growth and development [71]. Furthermore, hormone signal transduction is involved in plant growth regulation and root development under salt-stress conditions [72]. The KEGG pathway enrichment pathways of these differentially expressed genes were similar, and it may be that P. talassica × P. euphratica has a special response mechanism to salt stress.
Through the gene ontology functional enrichment analysis and the Kyoto Encyclopedia of Genes and Genomes pathway enrichment analysis, the metabolic pathways and molecular functions of the differentially expressed genes enriched in P. talassica × P. euphratica were clarified, and the response mechanism of salt stress in P. talassica × P. euphratica was revealed at the molecular level. A unique salt tolerance mechanism may provide a reference for further mining of salt tolerance genes of P. talassica × P. euphratica. Subsequent transgenic, proteomics and metabolomics studies should be carried out to explain the unique salt-tolerance mechanism of P. talassica × P. euphratica at multiple levels.

Reviewer 2 Report
Authors analyse the genes associated with the salt-stress response in P. tallasica x P. euphratica using RNA sequencing transcriptomic study. Research is generally well planned and performed. Conclusions are supported by obtained results. Some minor corrections should be included.
Line 134- lack of space after 1000
Line 136-138; how quickly the concentration of salt increased, until the final concentration was reached (200 or 400 mM) ?
Line 145- write precisely which standard method was used to isolate RNA.
Line 156- initial vol. and concentration of library.
Line 160- name of equipment and manufacturer
Line 161-162; rewrite the sentence.
Table 1 should be better formatted.
Line 352 remove doubled dot.
Authors could try to compare obtained results of genes that are potentially associated with the salt-stress response, with available transcriptomic or RT-PCR studies on P. tallasica, P. euphratica or related Populus sp. Several sentences related to this area with appropriate citations could be added to discussion section.
Author Response
Response to Reviewer 2 Comments
Point 1: Line 134- lack of space after 1000
Response 1: Thank you very much for your valuable comments, I have added spaces after 1000 as follows.
Modified sentence: The salt treatment group was treated with NaCl solution once every three days and the control group was irrigated with 1000 mL deionized water once every three days.
Point 2: Line 136-138; how quickly the concentration of salt increased, until the final concentration was reached (200 or 400 mM) ?
Response 2: Thank you very much for your valuable comments. I am sorry that the previous experimental design was not described in great detail. I have made corrections, I hope you can adopt it. The modified experimental design is as follows.
Modified sentence:
In order to avoid the salt shock effect, the potted P. talassica × P. euphratica seedlings were treated with salt stress in the way of increasing salt concentration until the predetermined concentration. In the 200 mmol/L NaCl treatment group, NaCl concentration increased by 50 mmol/L, 100 mmol/L, 150 mmol/L, and 200 mmol/L until the predetermined salt concentration of 200 mmol/L, and then treated with 200 mmol/L NaCl solution once. Keep the final salt concentration stable at 200 mmol/L. In the 400 mmol/L NaCl treatment group, NaCl concentration increased by 100 mmol/L, 200 mmol/L, 300 mmol/L, 400 mmol/L, until the predetermined concentration of 400 mmol/L, and then treated with 400 mmol/L NaCl solution once. Keep the final salt concentration stable at 400 mmol/L.
Point 3: Line 145- write precisely which standard method was used to isolate RNA.
Response 3: Thank you very much for your valuable comments. I have marked out the specific method, the specific method is as follows.
Modified sentence: RNA was extracted from tissues or cells using trizol extraction methods, followed by the rigorous quality control of the RNA samples using an Agilent 2100 BioAnalyzer and the accurate detection of RNA integrity.
Point 4: Line 156- initial vol. and concentration of library.
Response 4: Thank you very much for your valuable comments. I have marked the concentration of the library, the specific concentration is as follows.
Modified sentence: An Agilent 2100 BioAnalyzer was then used to determine the insert size of the library, and qRT-PCR was used to accurately quantify the effective library concentration (Effective library concentration is higher than 2nM) to ensure the library quality.
Point 5: Line 160- name of equipment and manufacturer
Response 5: Thank you very much for your valuable comments. I have marked the specific device name as follows.
Modified sentence: After qualified library inspection, Illumina sequencing was performed using Illumina MiSeq System (HiSeq 2500) after pooling different libraries in accordance with the effective concentration and the targeted offline data requirements.
Point 6: Line 161-162; rewrite the sentence.
Response 6: Thank you very much for your valuable comments. I have rewritten the sentence and the revised sentence is as follows.
Modified sentence: The basic principle of this sequencing technology is sequencing by synthesis.
Point 7: Table 1 should be better formatted.
Response 7: Thank you very much for your valuable comments. Table 1 I have made formatting changes in the revised manuscript.
Point 8: Line 352 remove doubled dot.
Response 8: Thank you very much for your valuable comments. I have removed the extra dots from the original.
Point 9: Authors could try to compare obtained results of genes that are potentially associated with the salt-stress response, with available transcriptomic or RT-PCR studies on P. tallasica, P. euphratica or related Populus sp. Several sentences related to this area with appropriate citations could be added to discussion section.
Response 9: Thank you very much for your valuable comments, which are very pertinent. I have made revisions to the discussion section to compare the results of differentially expressed genes that may be related to salt stress response. The specific content of the revision is as follows.
Modified sentence:
In the past few years, there has been an increasing number of studies using transcriptome sequencing to analyze salt tolerance in plants [58,59,60,61,62,63,64]. In this study, the salt tolerance of P. talassica × P. euphratica under different salt concentrations was comprehensively analyzed by RNA-seq.
Differentially expressed genes analysis results showed that the differentially expressed genes in roots, stems and leaves had different expression patterns. Overall, the following was true: the total number of differentially expressed genes in stems > the total number of differential genes in leaves > the total number of differential genes in roots. This is because differentially expressed genes with different expression patterns among roots, stems and leaves under salt-stress conditions were ideal targets for further functional research to understand the more specific molecular mechanisms of salt tolerance [62].
By screening the expression levels of differentially expressed genes, the possible salt tolerance genes in the roots, stems and leaves of P. talassica × P. euphratica were further explored. The highly expressed differential genes in each comparison group of leaves such as LOC105129452, LOC105129935 and LOC105114274 belong to the AP2/ERF transcription factor family. LOC105136804, LOC105124563 and LOC105124001 belong to the NAC (NAM,ATAF1,2,CUC) transcription factor family. LOC105131213, LOC105116811 and LOC105132264 belong to the WRKY transcription factor family. LOC105127881, LOC105142149, LOC105115176, LOC105141220, LOC105128502 and LOC105131234 belong to the bZIP transcription factor family. The highly expressed differential genes in each comparison group of stems such as LOC105119414, LOC105129607 and LOC105122741 belong to the AP2/ERF transcription factor family. LOC105142033, LOC105112164 and LOC105116962 belong to the NAC (NAM,ATAF1,2,CUC) transcription factor family. LOC105119522 , LOC105131256 and LOC105115491 belong to the WRKY transcription factor family. LOC105141088, LOC105141220, LOC105137881, LOC105126640, LOC105134677 and LOC105134493 belong to the bZIP transcription factor family. The highly expressed differential genes in each comparison group of roots such as LOC105108844, LOC105107217 and LOC105120059 belong to the AP2/ERF transcription factor family. LOC105135115, LOC105127231 and LOC105116469 belong to the NAC (NAM,ATAF1,2,CUC) transcription factor family. LOC105114369, LOC105124330 and LOC105127348 belong to the WRKY transcription factor family. LOC105134637, LOC105115176, LOC105115119, LOC105128502, LOC105112947 and LOC105124801 belong to the bZIP transcription factor family.
In response to abiotic and biotic stresses, plants often activate a battery of defense responses that include inducible expression of a set of stress-related genes, which are regulated directly or indirectly by different types of transcription factors [80]. Several classes of transcription factors, including AP2/ERF, bZIP, NAC (NAM, ATAF, and CUC) and WRKY have been reported to be associated with plant defense [100-103]. NAC (NAM, ATAF, and CUC) domain proteins are plant-specific transcriptional factors known to play diverse roles in plant growth and development [73,74]. NAC (NAM, ATAF, and CUC) transcription factors are also involved in plant responses to biotic and abiotic stress processes, including high salt [75], drought [76], freezing [77], and viral infection [78]. NAC (NAM, ATAF, and CUC) plant transcription factors regulate essential processes in development, stress responses and nutrient distribution in important crop and model plants (rice, Populus, Arabidopsis), which makes them play a huge role in crop improvement and production [79].
The AP2/ERF transcription factors are known to regulate diverse processes of plant development and stress responses. In addition, AP2/ERF transcription factors are ideal candidates for crop growth, development and improvement because their overexpression enhances tolerances to drought, salt stress in the transgenic plants [95]. A variety of AP2/ERF transcription factors have been successfully identified and investigated in some plants, including Arabidopsis, rice [81,82], poplar (Populus tricocarpa) [83], and wheat (Triticum aestivum) [84]. WRKY transcription factors, a family of regulatory genes, were first identified in plants [87–89]. WRKY transcription factors are key regulators of many plant processes, including the responses to biotic and abiotic stresses, senescence, seed dormancy and seed germination [85]. A large number of WRKY transcription factors have been reported from Arabidopsis, rice, and other higher plants [86]. The Populus genome contains at least 100 WRKY genes [91]. Increasing evidence has confirmed important roles of WRKY transcription factors in poplar defense processes, and several poplar WRKY genes have been identified to be involved in defense response [90].
The bZIP transcription factor gene family is one of the largest and most diverse families in plants. Current studies have shown that the bZIP proteins regulate numerous growth and developmental processes [92]. They have also been found to be involved in responses to a variety of abiotic/biotic stimuli, such as drought [93,94], and high salinity [95]. Some of these bZIP factors have been shown to confer disease resistance [96] and to trigger expression of defense-related genes in systemic acquired resistance (SAR) [97].The bZIP-type transcriptional factors are involved in developmental and physiological processes in response to stresses, and are important for various plants to withstand adverse environmental conditions [98,99].
Transcription factor families such as AP2/ERF, NAC (NAM, ATAF, and CUC), WRKY and bZIP are all stress-resistant transcription factors that have been reported to be involved in poplars and other plants sensing of early salt stress. The analysis found that these differential genes with high expression in the roots, stems and leaves of P. talassica × P. euphratica were also existed in its male parent P. euphratica. They are very likely to be the key salt-tolerant genes that exist in P. talassica × P. euphratica to resist external salt stress. In the future, the functions of these genes in P. talassica × P. euphratica and other tree species should be further analyzed, so as to provide some reference for the planting of P. talassica × P. euphratica in saline-alkali land.
A gene ontology functional enrichment analysis showed that, many differentially expressed genes were related to material metabolism, signal transmission and other processes, which was consistent with the annotation results of Reaumuria trigyna [65]. The genes involved in the above processes, such as signal transmission, ion transport, cell communication, cell periphery and cell wall, may be involved in P. talassica × P. euphratica responses to salt stress. Salt stress disturbs the normal growth and development of plants. Modifications of the cell wall are common defense responses when plants are suffering from abiotic and biotic stresses. In our study, most of the genes related to cell wall and growth were up-regulated under salt stress conditions [66]. A Kyoto Encyclopedia of Genes and Genomes pathway enrichment analysis showed that different genes were mainly enriched in Plant hormone signal transduction and MAPK-signaling pathway–plant, which were highly annotated, may be involved in P. talassica × P. euphratica responses to salt stress. This is similar to the research results of Zhan Jiang Han et al [58].
For plants under abiotic or biotic stress, signal transduction is very crucial for adjustment in such unfavorable conditions [67–70]. Plants respond to abiotic stresses by regulating complex signaling networks, which helps plants adapt to stress and thus enhance their growth and development [71]. Furthermore, hormone signal transduction is involved in plant growth regulation and root development under salt-stress conditions [72]. The KEGG pathway enrichment pathways of these differentially expressed genes were similar, and it may be that P. talassica × P. euphratica has a special response mechanism to salt stress.
Through the gene ontology functional enrichment analysis and the Kyoto Encyclopedia of Genes and Genomes pathway enrichment analysis, the metabolic pathways and molecular functions of the differentially expressed genes enriched in P. talassica × P. euphratica were clarified, and the response mechanism of salt stress in P. talassica × P. euphratica was revealed at the molecular level. A unique salt tolerance mechanism may provide a reference for further mining of salt tolerance genes of P. talassica × P. euphratica. Subsequent transgenic, proteomics and metabolomics studies should be carried out to explain the unique salt-tolerance mechanism of P. talassica × P. euphratica at multiple levels.

Round 2
Reviewer 1 Report
The discussion has improved a lot, however, the language still needs improvements.
Author Response
Point 1: The discussion has improved a lot, however, the language still needs improvements.
Response 1: Thank you very much for your valuable comments, I have found a professional teacher to revise the language of the article.
